# Loss of Nat4 and its associated histone H4 N-terminal acetylation mediates calorie restriction-induced longevity

Diego Molina-Serrano[1], Vassia Schiza[1], Christis Demosthenous[1], Emmanouil Stavrou[1], Jan Oppelt[2,3], Dimitris Kyriakou[1], Wei Liu[4], Gertrude Zisser[5], Helmut Bergler[5], Weiwei Dang[4] & Antonis Kirmizis[1,*]

## Abstract

Changes in histone modifications are an attractive model through which environmental signals, such as diet, could be integrated in the cell for regulating its lifespan. However, evidence linking dietary interventions with specific alterations in histone modifications that subsequently affect lifespan remains elusive. We show here that deletion of histone N-alpha-terminal acetyltransferase Nat4 and loss of its associated H4 N-terminal acetylation (N-acH4) extend yeast replicative lifespan. Notably, *nat4Δ*-induced longevity is epistatic to the effects of calorie restriction (CR). Consistent with this, (i) Nat4 expression is downregulated and the levels of N-acH4 within chromatin are reduced upon CR, (ii) constitutive expression of Nat4 and maintenance of N-acH4 levels reduces the extension of lifespan mediated by CR, and (iii) transcriptome analysis indicates that *nat4Δ* largely mimics the effects of CR, especially in the induction of stress-response genes. We further show that nicotinamidase Pnc1, which is typically upregulated under CR, is required for *nat4Δ*-mediated longevity. Collectively, these findings establish histone N-acH4 as a regulator of cellular lifespan that links CR to increased stress resistance and longevity.

**Keywords** calorie restriction; histone N-terminal acetylation; lifespan; Nat4; Pnc1

**Subject Categories** Ageing; Chromatin, Epigenetics, Genomics & Functional Genomics; Metabolism

## Introduction

The eukaryotic genome is packaged into chromatin, a macromolecular structure consisting of nucleosomes as the basic repeating unit. The nucleosomes themselves contain 147 bp of DNA wrapped around an octamer of histone proteins. Histones undergo several post-translational modifications that can induce changes in chromatin structure and thus regulate DNA-based processes like transcription [1]. In recent years, several studies in various organisms have demonstrated that changes in the abundance or distribution of post-translational modifications can alter chromatin structure during aging [2–10]. Therefore, alterations in histone modifications contribute to the transcriptional changes that are associated with aging [6,7,11–14]. Considering that these epigenetic marks respond to environmental stimuli [15], histone modifications could act as an interface through which extracellular signals, like dietary manipulations or stress, may impact on lifespan.

Calorie restriction (CR) is the most conserved and well-studied extracellular intervention that prolongs lifespan [16,17]. The budding yeast *Saccharomyces cerevisiae* has been a valuable aging model for elucidating conserved effects of CR [18,19], mainly through a regimen that reduces glucose concentration in growth media from 2 to 0.5% or lower [20]. Extensive studies in yeast that have used glucose limitation implicated multiple pathways in CR-mediated longevity. It was shown that CR suppresses rDNA instability and the formation of extra-chromosomal rDNA circles (ERCs) [21–25] as a result of increased Sir2 activity [26–28]. The function of Sir2 is stimulated in CR because of increased NAD+/NADH ratio [29] or through induction of nicotinamidase Pnc1, an important regulator of CR-mediated longevity that prevents the accumulation of intracellular nicotinamide (an inhibitor of Sir2) during times of stress [30,31]. Moreover, CR prolongs lifespan by repressing ribosome biogenesis and protein translation through downregulation of TOR signaling [32–34]. Finally, CR elicits lifespan extension by enhancing resistance to oxidative stress [35] and by inducing genotoxic stress response through inhibition of ATP-dependent chromatin remodeling [36]. Although CR-mediated pathways involve chromatin-based processes [37], there are no established links between CR and specific changes in histone modifications that subsequently affect longevity [38].

1 Department of Biological Sciences, University of Cyprus, Nicosia, Cyprus
2 CEITEC-Central European Institute of Technology, Masaryk University, Brno, Czech Republic
3 National Centre for Biomolecular Research, Masaryk University, Brno, Czech Republic
4 Huffington Center on Aging, Baylor College of Medicine, Houston, TX, USA
5 Institut für Molekulare Biowissenschaften, Karl-Franzens-Universität, Graz, Austria
*Corresponding author. Tel: +357 22 892678; E-mail: kirmizis@ucy.ac.cy

Nat4 (also known as NatD, Naa40, and Patt1) belongs to the family of N-terminal acetyltransferases (NATs), which catalyze the addition of an acetyl group to the primary alpha-amino group at the very N-terminal residue of a protein [39]. Protein N-terminal acetylation is one of the most abundant and conserved protein modifications, occurring in over 60% of eukaryotic proteins [40]. However, Nat4 is unique among other NAT enzymes because of its high substrate selectivity. So far, it is known to acetylate only the N-termini of histones H4 and H2A [39,41,42]. A handful of other proteins have been suggested to be targets of Nat4 [43,44], but whether this is correct still remains to be determined. The enzymatic activity of Nat4 toward H4 and H2A is conserved from yeast to human [41,45] and has been linked to transcriptional regulation [46,47]. More specifically, it was demonstrated that N-acH4 in yeast antagonizes the adjacent histone modification H4R3me2a to control ribosomal RNA expression. Importantly, the cross talk between N-acH4 and H4R3me2a responds to CR, suggesting that Nat4 and N-acH4 could act as a sensor for cell growth [46]. Consistent with that, studies in mice show that Naa40 controls lipid metabolism and fat mass [48] and studies in human cancer cells unveiled the role of this enzyme in apoptosis [47,49].

The previous evidence, which links N-acH4 to the rDNA locus and shows that its function responds to CR [46], has led us to hypothesize that H4 N-terminal acetylation and its associated enzyme are part of a mechanism that regulates lifespan. Accordingly, in this report, we demonstrate that Nat4 deletion (*nat4Δ*) and loss of N-acH4 extend yeast replicative lifespan through a pathway that mimics CR-mediated longevity. Consistently, we find that CR suppresses Nat4 expression resulting in reduced nucleosomal deposition of N-acH4 and constitutive expression of Nat4 limits the longevity effect of CR. Furthermore, we show that this pathway leading to extended lifespan involves the induction of specific stress-response genes and requires expression of nicotinamidase Pnc1. Overall, we present a model in which histone H4 N-terminal acetylation by Nat4 links CR to the induction of stress-response genes and longevity.

## Results

### Nat4 deletion extends lifespan through a CR-mediated pathway

We have previously shown that CR correlates with a reduction in the levels of H4 N-terminal acetylation (N-acH4) on *RDN25* [46]. To further explore whether this reduction is Nat4-dependent, we examined the expression levels of Nat4 during CR. In order to perform this analysis, we diminished the levels of glucose from 2% (no calorie restriction, NCR) to 0.1% (calorie restriction, CR) and harvested cells before glucose exhaustion and entry into stationary phase (Fig EV1A). We found that CR reduces significantly the expression of *NAT4* (Fig 1A). Accordingly, ChIP assays using an antibody against N-acH4 that was previously developed and characterized [46,47] show that in WT cells, the levels of N-acH4 on chromatin are strongly and significantly reduced across the rDNA region upon CR (Fig 1B, compare blue bars between NCR and CR). Notably, statistical analysis of *NAT4* expression using unpaired two-tailed Student's *t*-test showed that CR does not diminish further the levels of N-acH4 in cells already lacking Nat4 (*nat4Δ*), indicating that CR

controls this histone modification primarily through Nat4 (Fig 1B, compare red bars between NCR and CR). Importantly, *NAT4* expression is not significantly downregulated by CR when its transcription is constitutively driven by the CR-insensitive promoter *STE5*, confirming that endogenous *NAT4* is regulated by glucose deprivation (Fig 1A, see *Pste5-NAT4*). In line with this, constitutive expression of *NAT4* in the *Pste5-NAT4* strain maintains significantly higher N-acH4 levels at most rDNA loci when subjected to CR in comparison with WT cells (Fig 1B, compare green and blue bars in NCR and green versus blue bars in CR). Altogether, these results indicate that glucose limitation diminishes the levels of Nat4 and as a result reduces nucleosomal deposition of N-acH4.

Based on the fact that CR regulates the levels of Nat4 and since a link between CR and increased cellular lifespan has already been established [20,50], we hypothesized that Nat4 is part of a mechanism whereby CR extends cellular lifespan. To test this hypothesis, we examined the effect of Nat4 on yeast replicative lifespan (RLS) and observed that loss of Nat4 significantly extended lifespan approximately by 21% in a BY4741 background strain (Fig 1C). The extension of lifespan by *nat4Δ* was independent of the mating type and yeast strain background, as deletion of Nat4 in BY4742, YSC5106, and JK9-3Dα strains also extended lifespan by about 41, 26 and 20%, respectively (Fig 1D). To determine whether *nat4Δ* extends lifespan through a CR-mediated pathway, we examined replicative lifespan in *nat4Δ* cells that were simultaneously subjected to CR. As expected, we found that CR alone extended lifespan by about 31%, while *nat4Δ* was unable to extend lifespan further under these CR conditions (Fig 1E), suggesting that *nat4Δ* is epistatic to CR. To verify the connection between CR and *nat4Δ,* we also assayed lifespan epistasis in a *nat4Δ tor1Δ* double-mutant, since *tor1Δ* is a genetic mimic of CR [32]. As predicted, *nat4Δ tor1Δ* extended lifespan to the same degree as the *nat4Δ* single mutant alone (Fig EV1B). The above data support the idea that Nat4 depletion occurs downstream of a CR-activated pathway to confer lifespan extension. Consistent with this, constitutive expression of *NAT4* (Fig 1A) limits the extension of lifespan induced by CR from 32 to 19% (Fig 1F). Overall, our findings so far indicate that CR regulates the levels of Nat4 in order to extend replicative aging.

### Loss of Nat4 does not alter rDNA stability and polysome profiling

It was previously proposed that CR extends lifespan through mechanisms that either increase rDNA stability [24] or deplete 60S ribosomal subunits [33]. Since Nat4 is linked to rDNA regulation (Fig 1B and [46]), we next considered the possibility that *nat4Δ* extends lifespan through the above rDNA-dependent mechanisms. Silencing at the rDNA locus is mediated by the RENT (REgulator of Nucleolar silencing and Telophase exit) complex, consisting of the proteins Net1, Cdc14, and Sir2 [51]. Using a yeast strain in which Net1 was TAP-tagged, we initially found that *nat4Δ* significantly enhanced the association of the RENT complex along the rDNA locus compared to a WT strain (Fig 2A and B). This result suggested that *nat4Δ* may affect rDNA stability, since increased binding of the RENT complex had also been linked to enhanced rDNA stability and longevity [23]. Thus, we next monitored rDNA copy number in young and old cells using quantitative real-time PCR. As expected, we detected increased rDNA copy number in WT old cells compared to young cells (Fig 2C) due to accumulation of ERCs, which constitute an

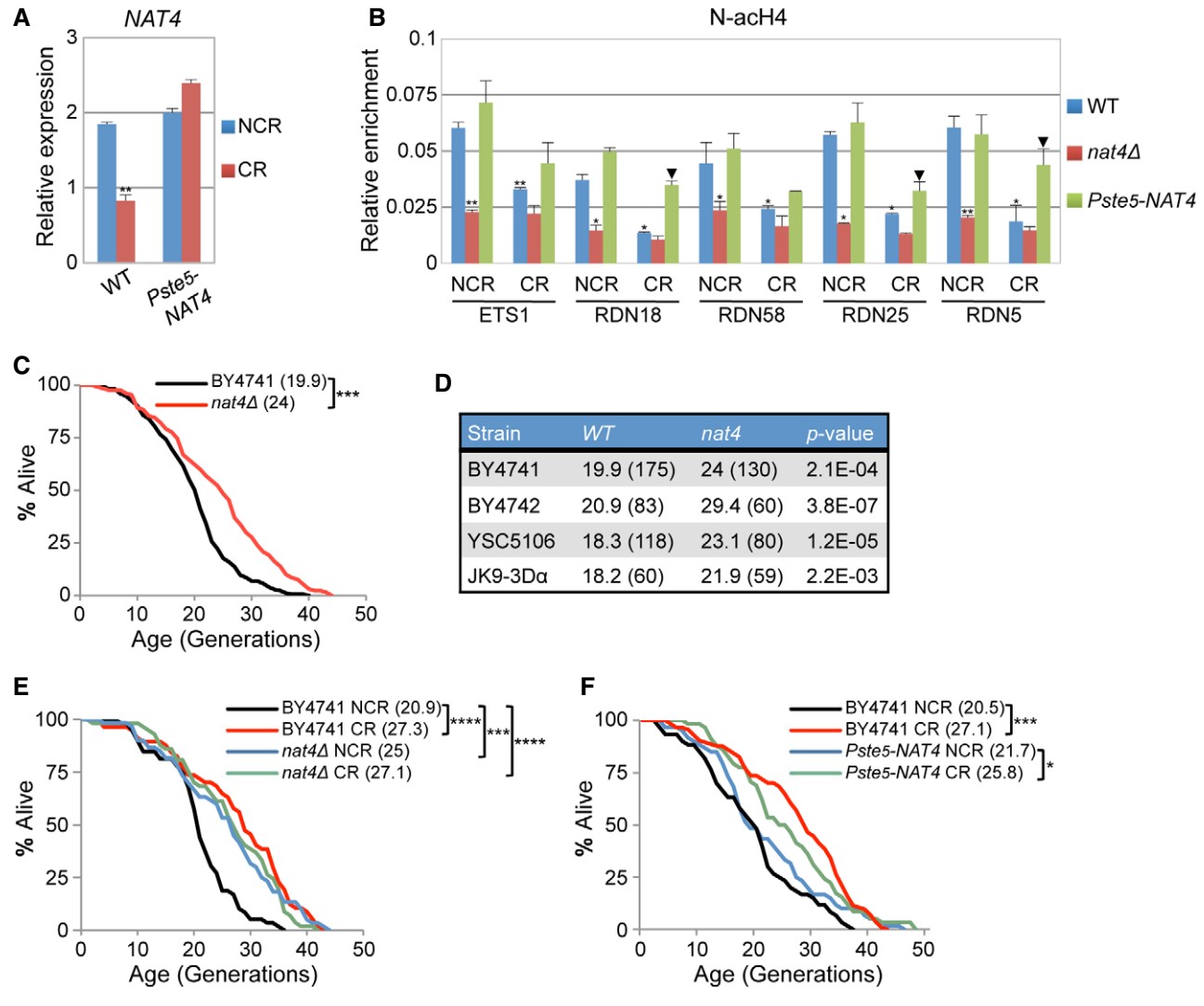

**Figure 1. NAT4 deletion extends lifespan through a calorie restriction-mediated pathway.**

A   Expression levels of *NAT4* analyzed by qRT–PCR using total RNA extracted from a wild-type (BY4741) and *Pste5-NAT4* strain grown in 2% (NCR), or 0.1% glucose (CR). *NAT4* levels were normalized to *TAF10*, whose expression remains unchanged. Error bars, SEM (standard error of the mean) of three independent experiments. **$P \leq 0.01$; calculated by unpaired two-tailed Student's *t*-test.

B   ChIP analysis performed in BY4741 wild-type, *nat4Δ*, and *Pste5-NAT4* strains grown in the same conditions as in (A). Chromatin was immunoprecipitated using an antibody against N-acH4 and analyzed by qRT–PCR using primers for the different rDNA regions. Enrichment was normalized to histone H4 levels. Error bars, SEM of three independent experiments. Statistical significance was determined by unpaired two-tailed Student's *t*-test: *$P \leq 0.05$; **$P \leq 0.01$ compared values to wild-type NCR conditions; arrowhead $P \leq 0.05$ compared values to wild-type CR conditions.

C   Replicative lifespan (RLS) for BY4741 wild-type and *nat4Δ* strains. Values in parentheses (here and hereafter) indicate mean lifespan.

D   Mean RLS for wild-type and *nat4Δ* strains using different genetic backgrounds. Values outside of parentheses indicate mean lifespan, while values in parentheses indicate the number of cells examined.

E   Mean RLS for BY4741 wild-type and *nat4Δ* strains under NCR and CR conditions.

F   Mean RLS for BY4741 wild-type and *Pste5-NAT4* strains under NCR and CR conditions.

Data information: (C-F) Statistical significance was determined by one-way ANOVA test: *$P \leq 0.05$; **$P \leq 0.01$; ***$P \leq 0.001$; ****$P \leq 0.0001$.

important cause of aging in yeast [36]. Furthermore, the absence of the replication fork protein Fob1 resulted in reduced rDNA copy number in *fob1Δ* old cells compared to WT old cells, which is consistent with previous reports showing that *fob1Δ* stimulates rDNA stability to promote longevity [52,53]. Notably, deletion of *NAT4* did not influence the accumulation of rDNA copy number in old cells compared to a WT strain (Fig 2C), indicating that *nat4Δ* does not alter recombination at the rDNA locus.

It has been previously reported that reduced ribosomal 60S subunit abundance and thus mRNA translation extend yeast replicative lifespan [33]. To determine whether the stoichiometry of ribosomal subunits and protein translation are affected in *nat4Δ*-induced longevity, we generated polysome profiles for *nat4Δ* and its corresponding WT strain. We observed no differences in the formation of free ribosomal subunits (40S, 60S, and 80S) or polyribosome peaks between the two strains (Fig 2D). Therefore, despite the fact that

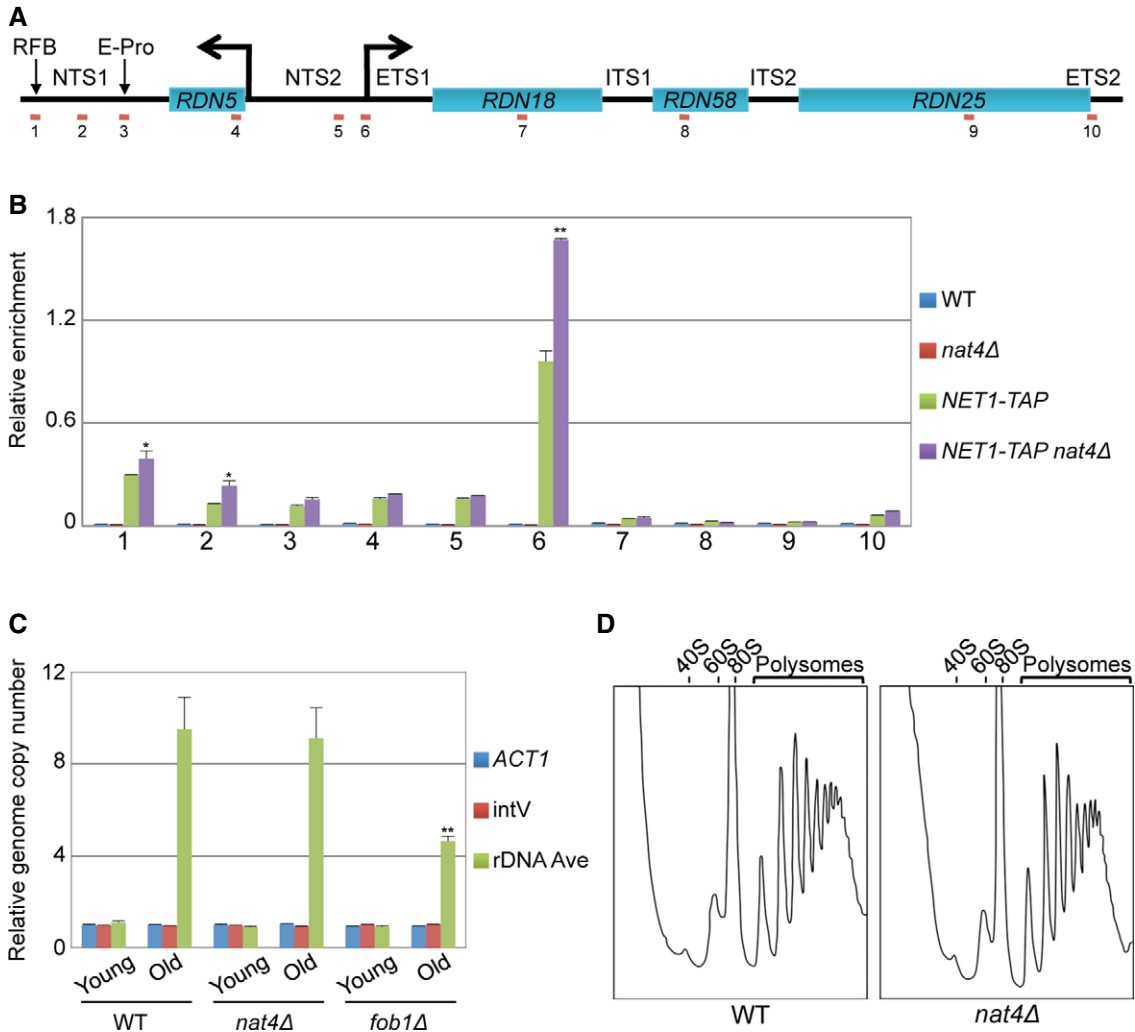

**Figure 2.    rDNA stability and mRNA translation are unaffected in nat4Δ.**

A   Schematic of the budding yeast rDNA locus on chromosome XII. Primers along the rDNA locus are indicated by red lines and numbers 1 to 10. NTS (non-transcribed spacer), ETS (external transcribed spacer), ITS (internal transcribed spacer), RFB (replication fork barrier), E-pro (bidirectional promoter).

B   ChIP experiments were performed in SF10 wild-type, nat4Δ, NET1-TAP, and NET1-TAP nat4Δ strains using an anti-IgG antibody. The enrichment from the antibody was normalized to input. Error bars, SEM of three independent experiments. Statistical significance was determined by unpaired two-tailed Student's t-test: **$P \leq 0.01$ compared nat4Δ to WT values. *$P \leq 0.05$.

C   Quantitative real-time PCR analysis of rDNA copy number for young and old cells in BY4741 wild-type, nat4Δ, and fob1Δ strains. Error bars, SEM of three independent experiments. Statistical significance was determined by unpaired two-tailed Student's t-test: **$P \leq 0.01$ compared to corresponding WT values.

D   Polysome profiling analysis performed in BY4741 wild-type and nat4Δ strains. The plots are representative of two independent experiments.

loss of Nat4 reduces the abundance of all rRNAs [46], the composition of ribosomes and mRNA translation is not affected in nat4Δ. In total, the above findings indicate that nat4Δ-induced longevity is mediated through a mechanism that is independent of rDNA stability and translation.

### NAT4 deletion mimics CR in inducing a cohort of stress-response genes

Having ruled out a link between the ribosomal DNA locus and nat4Δ-induced longevity, we turned our attention to the identification of other Nat4-regulated genes. To determine which genes are regulated by Nat4, we performed RNA-seq in nat4Δ and isogenic

wild-type strains. We identified 138 upregulated and 59 downregulated genes after applying a cutoff of twofold change in either direction (Tables EV1 and EV2). Gene ontology (GO) clustering analysis for all deregulated genes showed characteristic changes expected for cells in CR [36,54,55]. Specifically, the enriched GO terms included upregulated protein and carbohydrate metabolic processes, respiration, and phosphorylation as well as downregulated ribonucleoprotein complex biogenesis (Fig 3A). Notably, the most significantly enriched GO cluster among upregulated genes in nat4Δ was the response to abiotic stimulus, which is a term that encompasses various forms of cellular stress response (Fig 3A). To verify the similarity in transcriptome changes between nat4Δ and CR, we performed RNA-seq analysis of WT cells grown in 0.1% glucose (CR)

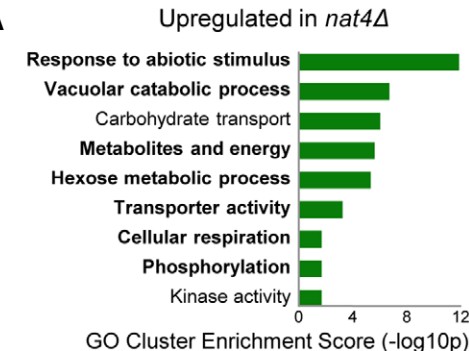

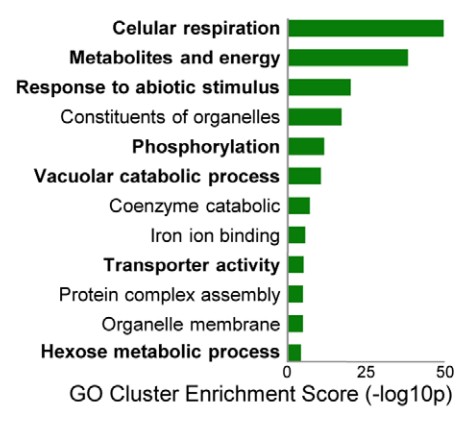

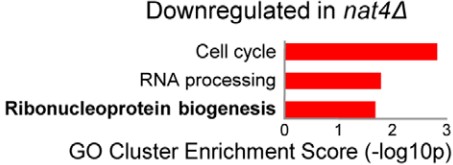

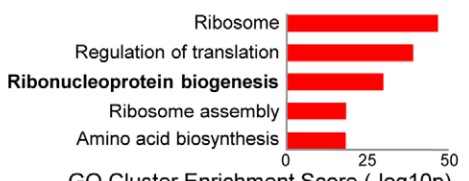

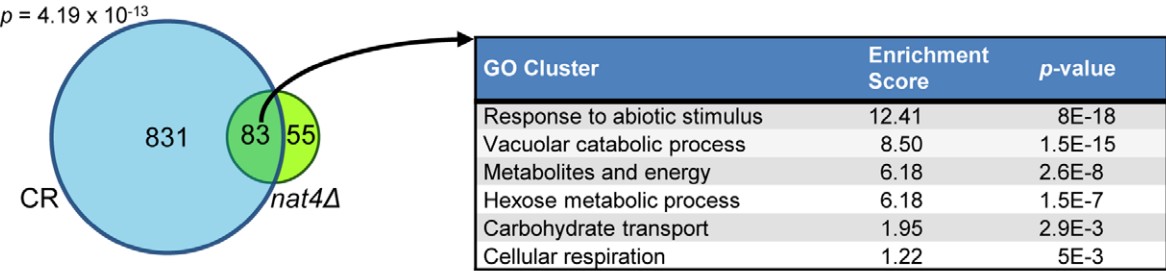

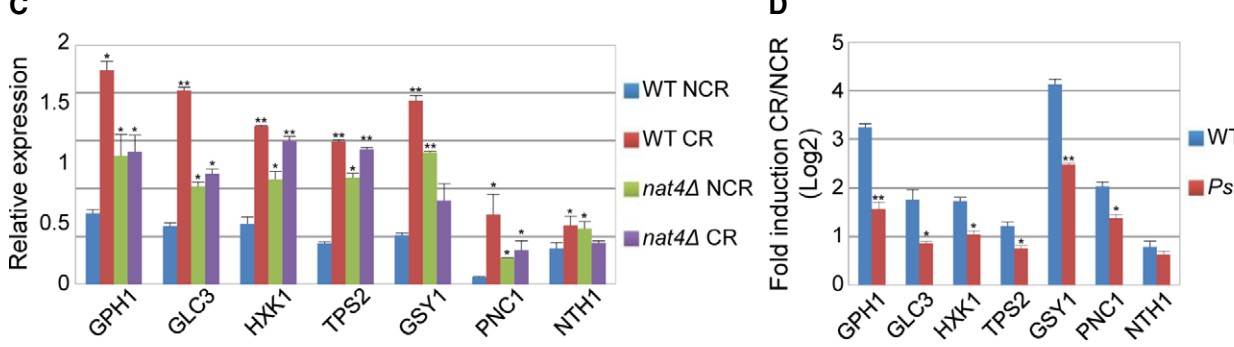

**Figure 3. *nat4Δ* mimics transcriptome effects of calorie restriction.**

A   Clustering analysis for upregulated and downregulated genes in BY4741 *nat4Δ* (NCR, 2% glucose) and wild-type (CR, 0.1% glucose) strains. Gene ontology was calculated by DAVID software (version 6). GO categories with enrichment scores of *P* < 0.05 are shown. Common GO cluster terms between nat4-delta and CR are shown in bold.

B   Venn diagram showing statistically significant overlap (83 genes) between upregulated genes in *nat4Δ* cells (138) and wild-type cells grown in CR conditions (914). Significance of overlap between the two sets is shown (*P* = 4.19 × 10⁻¹³; calculated using hypergeometric test [phyper]). The most relevant GO categories from the overlapping genes are indicated in the table on the right.

C   Gene expression analysis of selected upregulated stress-response genes (according to results in 3A) in BY4741 wild-type and *nat4Δ* strains grown under NCR and CR conditions. Expression levels were normalized to *TAF10*, whose expression remains unchanged. Error bars, SEM of three independent experiments. *P*-values were obtained comparing results to wild-type under NCR conditions and calculated by unpaired two-tailed Student's *t*-test: *$P ≤ 0.05$; **$P ≤ 0.01$.

D   Gene expression analysis showing fold induction levels (Log2) of selected genes comparing BY4741 wild-type and *STE5p-NAT4* strains under NCR (2% glucose) and CR (0.1% glucose) conditions. Expression levels were normalized to *TAF10*, whose expression remains unchanged. Error bars, SEM of three independent experiments. *P*-values were calculated by unpaired two-tailed Student's *t*-test: *$P ≤ 0.05$; **$P ≤ 0.01$.

compared to control cells grown in 2% glucose. GO clustering analysis validated our previous observations, since 8 out of 12 enriched GO cluster terms in *nat4Δ* were also enriched in CR (Fig 3A, compare left with right plots).

Since similar GO cluster terms were enriched in both *nat4Δ* and CR-treated cells, we compared the list of upregulated and downregulated genes from the two datasets. Among all upregulated genes, there was a statistically significant overlap (83 genes, Table EV3) between those upregulated in *nat4Δ* (138 genes) and in CR (914 genes) (Fig 3B, Venn diagram). Remarkably, the commonly upregulated genes enriched almost all GO terms that were similar between *nat4Δ* and CR above (Fig 3A), with response to abiotic stimulus being the most enriched (Fig 3B, table). On the other hand, from all the downregulated genes, only six overlapped between the two conditions (Fig EV2 and Table EV3). These data indicate that *NAT4* deletion partially resembles CR upregulation at the transcriptome level, especially in the induction of stress-response genes.

To further demonstrate that *nat4Δ* and CR control the expression of certain genes in a similar manner, we examined the expression of seven stress-response genes that were commonly upregulated in the two RNA-seq datasets using RT–qPCR. We found that the expression of all seven stress-response genes (*GPH1, GLC3, HXK1, TPS2, GSY1, PNC1,* and *NTH1*) is upregulated in *nat4Δ* and CR-treated cells (Fig 3C), thus validating the RNA-seq findings. Notably, there was no additive upregulation of these seven genes when *nat4Δ* cells were concomitantly grown in CR conditions (Fig 3C, see *nat4Δ* CR), consistent with the epistasis between *nat4Δ* and CR observed during the RLS assays (Figs 1E and EV1). Moreover, constitutive *NAT4* expression significantly reduces the induction of these stress-response genes by CR (Fig 3D), demonstrating that Nat4 is a key regulator downstream of a CR-mediated pathway, as also indicated by the lifespan experiments above (Fig 1F). Taken together, these results show that *nat4Δ* and CR deregulate a similar set of target genes and propose that Nat4 acts downstream of CR to control the expression of stress-response genes.

### *nat4Δ*-induced longevity is mediated by loss of H4 N-terminal acetylation

We next sought to investigate whether *nat4Δ*-induced longevity was dependent on its acetyltransferase activity toward histones. We first created a Nat4 catalytic mutant in which the strictly conserved glutamic acid 186 among eukaryotic Nat4 proteins (Fig EV3A) was converted to alanine (E186A). This residue was previously shown to be critical for the enzymatic activity of the mammalian Nat4 orthologs [43,49]. To confirm that Nat4-E186A mutant was inactive in yeast cells, we examined the levels of H4R3me2a whose occurrence is inhibited by Nat4-mediated H4 N-terminal acetylation [46]. We detected robust accumulation of H4R3me2a in the Nat4-E186A strain that mimics the effect of *nat4Δ*, demonstrating that the acetyltransferase activity is abolished in the E186A mutant (Fig EV3B). Replicative lifespan experiments using the Nat4-E186A catalytic mutant strain showed an extension in lifespan of about 17% compared to an isogenic WT control strain (Fig 4A), indicating that loss of Nat4 acetyltransferase activity is sufficient to promote longevity.

Next, we wanted to determine whether histone N-terminal acetylation mediated by Nat4 is implicated in *nat4Δ*-induced longevity and stress response. Between the two known histone substrates of

Nat4 [41], we decided to focus on histone H4 because in a previous study we have shown that H4 is the main target through which Nat4 regulates rDNA expression [46]. Therefore, to do this, we used a yeast strain that expresses endogenous histone H4 with serine 1 mutated to aspartate (H4S1D) or alanine (H4S1A). Lifespan assays showed that H4S1D and H4S1A largely mimic the effect of *nat4Δ*, extending lifespan by ~33 and 21%, respectively (Figs 4B and EV3C).

The above result prompted us to examine whether loss of H4 N-terminal acetylation leads to the induction of stress-response genes. We found that in the H4S1D and H4S1A strains, almost all stress-response genes examined had a statistically significant increase in their expression compared to a WT control strain (Figs 4C and EV3D), an induction that resembles the increase observed when *NAT4* was deleted in this strain background (Fig 4C). Finally, to verify the link of N-acH4 with the expression of these stress-response genes, we monitored their gene expression levels in a *nat4Δ* strain that expresses ectopically the human ortholog of Nat4 (hNaa40) (Fig EV3E). It was previously shown that ectopic expression of hNaa40 in yeast cells lacking Nat4 restores N-terminal acetylation of H4 but not of H2A [45]. In agreement with our data above, we found that expression of hNaa40 in a *nat4Δ* strain restores the expression of stress-response genes down to WT levels (Fig 4D). Overall, these results show that loss of H4 N-terminal acetylation can extend lifespan and induce the expression of stress-response genes, and therefore, its absence mimics the longevity effect of *nat4Δ*.

### H4R3 is required for *nat4Δ*-mediated longevity and induction of stress-response genes

We reported recently that methylation at H4R3 is necessary and sufficient for the regulation of rDNA expression by Nat4 [46]. Therefore, we were wondering whether this residue, H4R3, also regulates the effect of *nat4Δ* on longevity and induction of stress-response genes. We initially examined replicative lifespan in a *nat4Δ* strain that expresses histone H4 carrying a mutation at arginine 3 to lysine (H4R3K), preventing its methylation [46]. Notably, we found that the H4R3K mutation blocks the extension of lifespan that is normally promoted by *nat4Δ* (Fig 5A, compare *nat4Δ* to *nat4Δ* H4R3K). In fact, the H4R3K mutant had an opposite effect on lifespan compared to *nat4Δ* (Fig 5A), H4S1D (Fig 4B), and H4S1A (Fig EV3C), since it shortened lifespan by about 40%. This is consistent with the antagonistic relationship between N-acH4 and H4R3me2a that was previously described [46]. Next, we looked at the expression of stress-response genes in the presence of the H4R3K mutant. We found that H4R3K blocks the induction of stress-related genes mediated by *nat4Δ* since in the double-mutant strain *nat4Δ* H4R3K, the RNA levels of the tested genes were comparable to those obtained in a WT strain (Fig 5B). Taken together, these data show that H4R3 and most likely its methylation are part of the mechanism by which Nat4 controls expression of stress-response genes and cellular lifespan.

### Deletion of *NAT4* potentiates stress response in young and old cells

To investigate further the contribution of *nat4Δ* to the stress response associated with aging, we compared gene expression

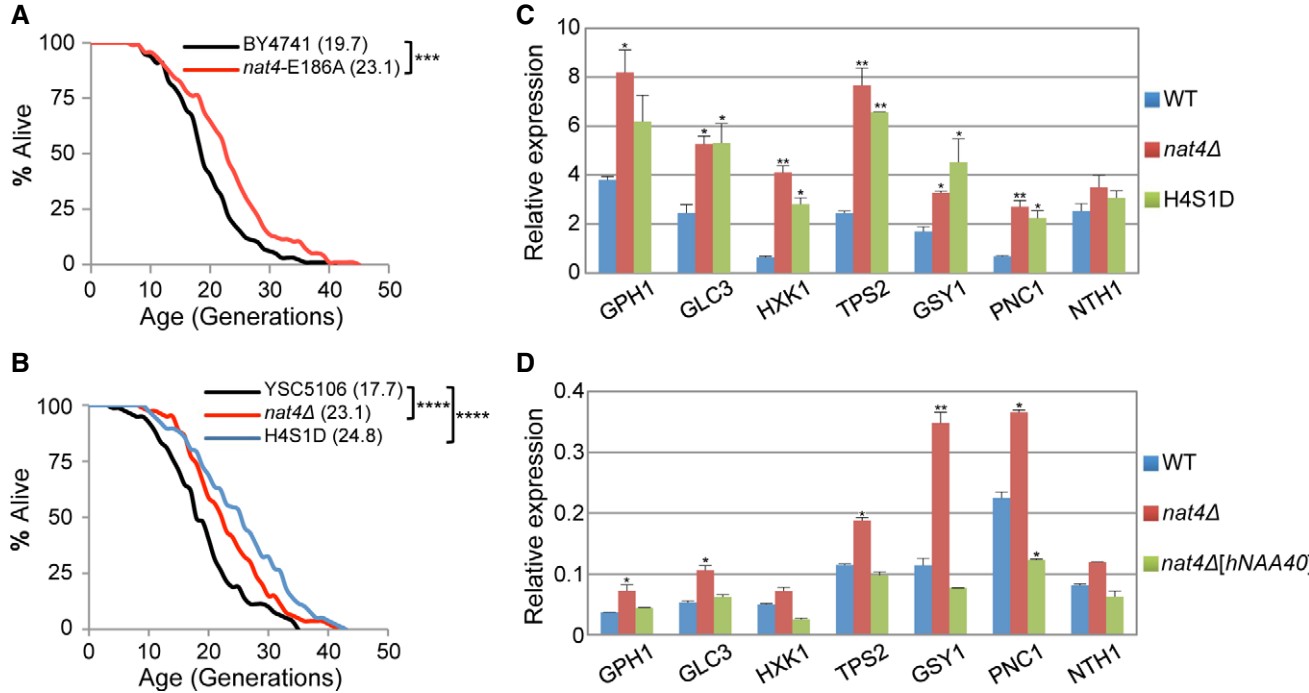

**Figure 4.**  *nat4Δ*-induced longevity is mediated through loss of H4 N-terminal acetylation.

A, B  (A) RLS analysis for BY4741 wild-type and *nat4-E186A* strains. (B) Mean RLS for YSC5106 wild-type, *nat4Δ*, and H4S1D strains. Values in parentheses indicate mean lifespan. Statistical significance was determined by one-way ANOVA test: ***$P \leq 0.001$; ****$P \leq 0.0001$.

C  Gene expression analysis of the indicated stress-induced genes in YSC5106 wild-type, *nat4Δ*, and H4S1D strains. Expression levels were normalized to *RPP0* whose expression remains unchanged. Error bars, SEM of three independent experiments. *P*-values were obtained by comparing *nat4Δ* and H4S1D to wild-type values and calculated by unpaired two-tailed Student's *t*-test: *$P \leq 0.05$; **$P \leq 0.01$.

D  Gene expression analysis of the indicated stress-induced genes in BY4742 wild-type, *nat4Δ*, and *nat4Δ[hNAA40]* strains. Expression levels were normalized to *ACT1*, whose expression remains unchanged. Error bars, SEM of three independent experiments. Statistical significance was determined by unpaired two-tailed Student's *t*-test: *$P \leq 0.05$ and **$P \leq 0.01$ compared to corresponding wild-type values.

changes for stress-response genes in young and old cells isolated from WT and *nat4Δ* strains. We found that in WT cells, a number of stress-response genes were upregulated in old cells compared to the young population (Fig 6A), which is expected since overall stress is elevated in aged cells [36]. Accordingly, *NAT4* expression levels were reduced in old versus young WT cells (Fig 6A), which is in agreement with the above transcriptome data showing that lack of Nat4 induces the expression of stress-response genes (Fig 3).

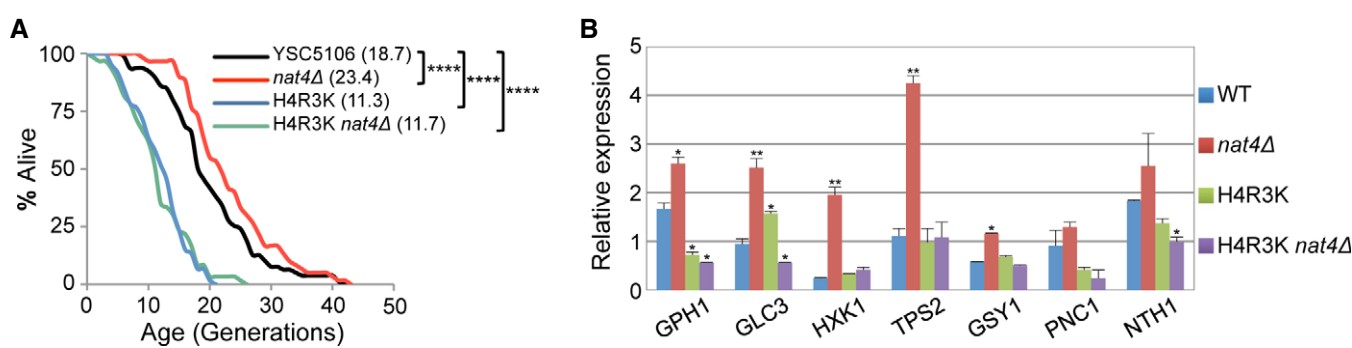

**Figure 5.**  H4R3 is required for *nat4Δ*-induced longevity and activation of stress-response genes.

A  RLS analysis for YSC5106 wild-type, *nat4Δ*, H4R3K, and double-mutant strains. Values in parentheses indicate mean lifespan. Statistical significance was determined by one-way ANOVA test: ****$P \leq 0.0001$.

B  Gene expression analysis of the indicated stress-induced genes in YSC5106 WT, *nat4Δ*, H4R3K, and double-mutant strains. Expression levels were normalized to *RPP0*, whose expression remains unchanged. Error bars, SEM of three independent experiments. Statistical significance was determined by unpaired two-tailed Student's *t*-test: *$P \leq 0.05$ and **$P \leq 0.01$ compared to corresponding wild-type values.

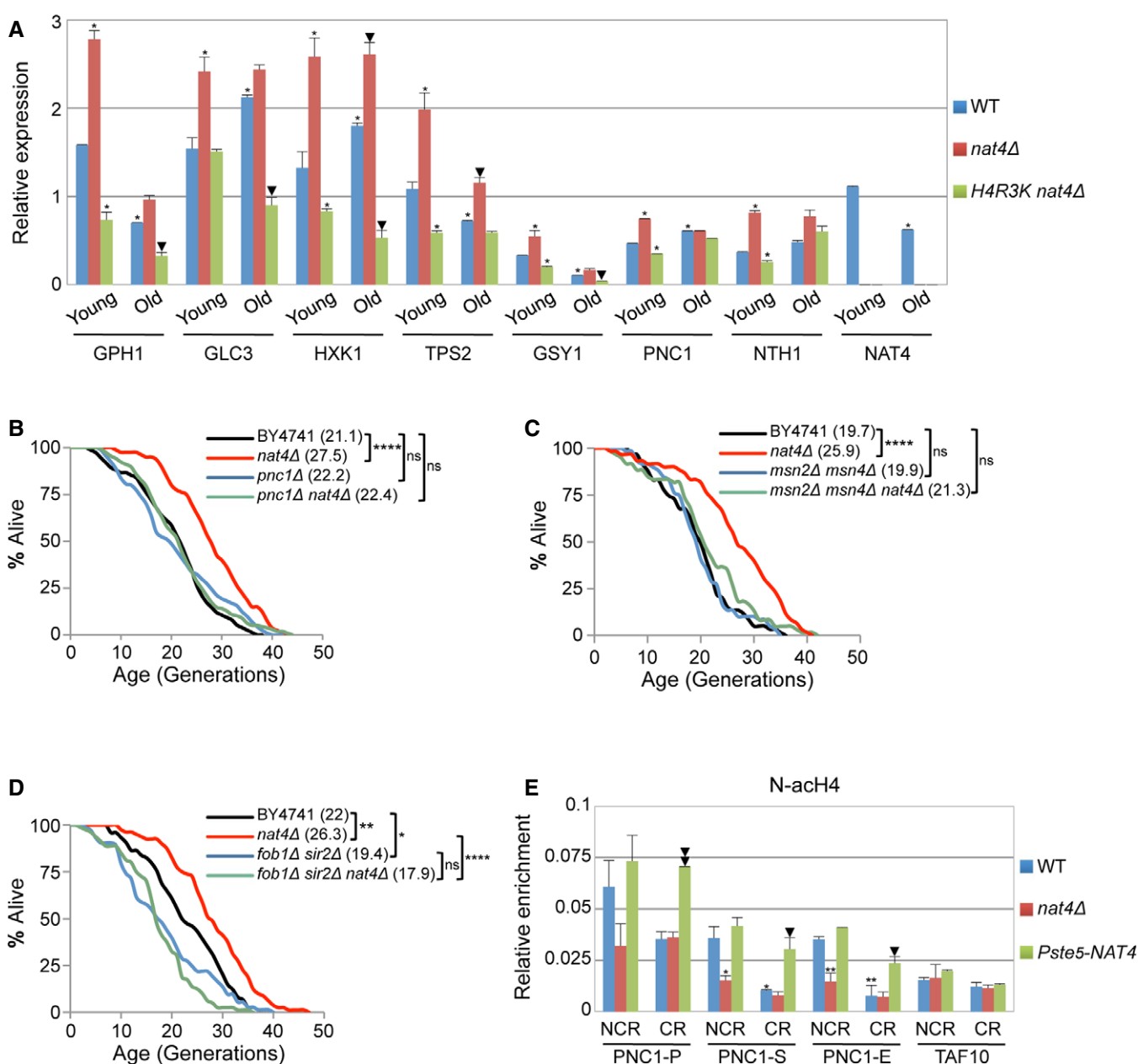

**Figure 6. *PNC1* is required for lifespan extension by *nat4Δ*.**

A   Gene expression analysis of the indicated stress-induced genes in young and old cells isolated from YSC5106 WT, *nat4Δ*, and H4R3K *nat4Δ* strains. Expression levels were normalized to *ACT1*, whose expression remains unchanged. Error bars, SEM of three independent experiments. *P*-values were calculated by unpaired two-tailed Student's *t*-test: **P* ≤ 0.05 compared to young wild-type; arrowhead *P* ≤ 0.05 compared to old wild-type.

B–D   (B) RLS analysis for WT, *nat4Δ*, *pnc1Δ*, and double-mutant strains in BY4741 genetic background. (C) Mean RLS in BY4741 wild-type, *msn2Δ msn4Δ* double-, and *msn2Δ msn4Δ nat4Δ* triple-mutant strains. (D) Mean RLS in BY4741 wild-type, *nat4Δ*, *fob1Δ sir2Δ* double-, and *fob1Δ sir2Δ* nat4Δ triple-mutant strains. Values in parentheses indicate mean lifespan. Statistical significance was determined by one-way ANOVA test: **P* ≤ 0.05; ***P* ≤ 0.01; ****P* ≤ 0.001; *****P* ≤ 0.0001. ns = non-significant.

E   ChIP analysis performed in BY4741 wild-type, *nat4Δ*, and *Pste5-NAT4* strains grown under NCR and CR conditions. Chromatin was immunoprecipitated using antibodies against N-acH4 and histone H4 and analyzed by qRT–PCR with the indicated primers. PNC1-P (promoter), PNC1-S (transcriptional start), PNC1-E (end of gene). The enrichment from the antibody was normalized to histone H4. Error bars, SEM of three independent experiments. Statistical significance was determined by unpaired two-tailed Student's *t*-test: **P* ≤ 0.05 and ***P* ≤ 0.01 compared values to wild-type NCR conditions; arrowhead *P* ≤ 0.05 and double arrowhead *P* ≤ 0.01 compared values to wild-type CR conditions.

Remarkably, the expression of stress-response genes in the young and old populations was much greater in the *nat4Δ* cells compared to the isogenic WT strain (Fig 6A). This enhanced expression in

young and old *nat4Δ* cells is abolished when the H4R3K mutant is introduced in the *nat4Δ* strain, demonstrating once again the requirement of H4R3 in *nat4Δ*-mediated effects. Altogether, these

findings propose that Nat4 deletion leads to a moderate increase in stress resistance early on in the life of a cell that is established by enhanced expression of stress-response genes. Such elevated stress resistance could prolong lifespan by allowing cells to more effectively antagonize age-associated cellular stress. This type of moderate increase in stress resistance is termed hormesis and was previously linked to lifespan extension [36,56,57].

### *PNC1* is essential for *nat4Δ*-mediated longevity

One of the significantly induced stress-response genes in *nat4Δ*, H4S1D, and H4S1A is *PNC1* (Figs 3D, 4C and EV3D), which encodes a nicotinamidase responsible for converting nicotinamide to nicotinic acid [58]. Pnc1 is an important regulator of CR-mediated longevity [30], which prolongs lifespan by preventing the accumulation of intracellular nicotinamide during times of stress [31]. Based on the above evidence, we sought to determine whether Pnc1 induction was necessary for *nat4Δ*-mediated longevity. In accordance with previous findings, *PNC1* deletion alone did not alter replicative lifespan relative to a WT strain (Fig 6B and [30]). Strikingly, when *PNC1* was deleted within *nat4Δ* cells, it decreased their lifespan back to WT levels (Fig 6B). Since the stress-responsive transcription factors Msn2 and Msn4 have been previously shown to directly activate the expression of *PNC1* [30,59], we also examined replicative lifespan in a *nat4Δ* strain lacking Msn2 and Msn4. Initially, we validated that in the triple-mutant *nat4Δ msn2Δ msn4Δ,* the expression of Pnc1 is not induced compared to WT cells (Fig EV4A), but in fact it is downregulated as previously demonstrated for the *msn2Δ msn4Δ* double-mutant strain [59]. Consistent with the results in Fig 6B, the absence of Pnc1 induction prevented *nat4Δ*-mediated lifespan extension in the triple-mutant strain *nat4Δ msn2Δ msn4Δ* (Fig 6C). Altogether, these results support the conclusion that overexpression of Pnc1 is stimulated by loss of Nat4 (Figs 3C, 4C and 6A, and Table EV1) and is essential for *nat4Δ*-mediated longevity.

Previous studies proposed that overexpression of Pnc1 limits the accumulation of intracellular nicotinamide and, as a result, induces longevity by stimulating Sir2 activity [30,31]. This conclusion was supported by an epistasis experiment showing that deletion of Sir2 in a PSY316 background strain overexpressing Pnc1 (*5XPNC1*) results in short lifespan similar to that observed for the *sir2Δ* single mutant [30,31]. However, interpreting longevity epistasis data in short-lived *sir2Δ* background strains have been recently challenged [60]. Therefore, we then sought to determine whether lifespan extension by Pnc1 upregulation is indeed dependent on Sir2 by investigating longevity in a *sir2Δ fob1Δ* double-mutant strain that overexpresses *PNC1* (*5XPNC1 sir2Δ fob1Δ*). We found that Pnc1 overexpression failed to extend lifespan in the *sir2Δ fob1Δ* PSY316 background strain, suggesting that Pnc1-mediated longevity is dependent on Sir2 (Fig EV4B). Although this result may appear inconsistent with a previous study which has shown that overexpression of Sir2 in the PSY316 strain background does not extend lifespan [61], it remains possible that Sir2 overexpression does not recapitulate Sir2 activation mediated through Pnc1 overexpression in this yeast background strain. This notion is also supported by the transcriptome data above, because we did not detect upregulation of Sir2 in the *nat4Δ* strain (Fig 3A) even though these single-mutant cells induce Pnc1 expression (Fig 3C) and have increased lifespan (Fig 1C). Moreover, due to the above link between Pnc1 and Nat4,

we then checked whether *nat4Δ*-mediated longevity is also dependent on Sir2 by examining replicative lifespan of the triple-mutant *nat4Δ sir2Δ fob1Δ* strain. Similar to Pnc1 overexpression, we found that *nat4Δ* fails to increase lifespan in the absence of Sir2 and Fob1 (Fig 6D), thus providing further evidence that Nat4 and Pnc1 function within the same longevity pathway.

Based on the transcriptome data above (Fig 3 and Table EV3) and previous reports [30,59], CR stimulates the expression of Pnc1. To further explore the link between calorie restriction, Nat4 and Pnc1, we then monitored the levels of N-acH4 on the *PNC1* gene under CR conditions. Using ChIP analysis, we detected significantly reduced levels of N-acH4 across the *PNC1* locus during CR but no overall change in the *TAF10* control locus (Fig 6E), demonstrating that increased Pnc1 expression by CR coincides with a reduction of N-acH4. As previously shown in Fig 1B, statistical analysis indicates that CR does not diminish further the levels of N-acH4 in cells already lacking Nat4 (*nat4Δ*), illustrating once again that CR controls N-acH4 predominantly through Nat4 (Fig 6E, compare red bars between NCR and CR). Remarkably, constitutive *NAT4* expression (*Pste5-NAT4*) under CR limits the removal of N-acH4 from chromatin in comparison with WT cells (Fig 6E, compare green and blue bars in NCR and green versus blue bars in CR), which is similar to what has been observed at most rDNA loci above (Fig 1B). This result is in agreement with the reduced induction of *PNC1* by CR when *NAT4* is maintained active (Fig 3D) and reinforces the notion that loss of N-acH4 by *nat4Δ*, H4S1D, or H4S1A mutation is necessary to induce Pnc1 expression (Figs 3C, 4C and EV3D, respectively). Additionally, we show that CR reduces the levels of N-acH4 at the promoter of six other *nat4Δ*-induced genes (Fig EV5), linking this histone modification to the induction of several stress-response genes. Overall, these findings suggest that reduced deposition of N-acH4 onto chromatin, either by CR or by Nat4 depletion, is associated with enhanced expression of *PNC1* and of other stress-response genes, supporting the conclusion that Nat4 functions downstream of a CR-mediated pathway to regulate gene expression and cellular lifespan.

## Discussion

Here, we identify Nat4 and its associated histone H4 N-terminal acetylation as novel regulators of yeast replicative lifespan. They function within a CR-controlled pathway, and their absence leads to the induction of stress response at the transcriptional level. To our knowledge, this constitutes one of the first examples where dietary intervention modulates a specific histone modification, which in turn directly impacts on the expression of certain genes that affect cellular lifespan. Hence, this study establishes N-acH4 as a molecular link between changes in chromatin that are induced by diet to control cellular lifespan (Fig 7).

We have previously shown N-acH4 cross talks with an adjacent H4 modification on arginine 3 in yeast. Specifically, N-acH4 inhibits the deposition of H4R3 asymmetric dimethylation to regulate rDNA silencing, as evidenced by enhanced H4R3me2a levels in the absence of Nat4 acetyltransferase activity (Fig EV3B and [46]). Consistent with this modification cross talk, we show that the H4R3 residue is required for *nat4Δ*-induced longevity and upregulation of stress-response genes (Figs 5 and 6A), implicating its methylation in

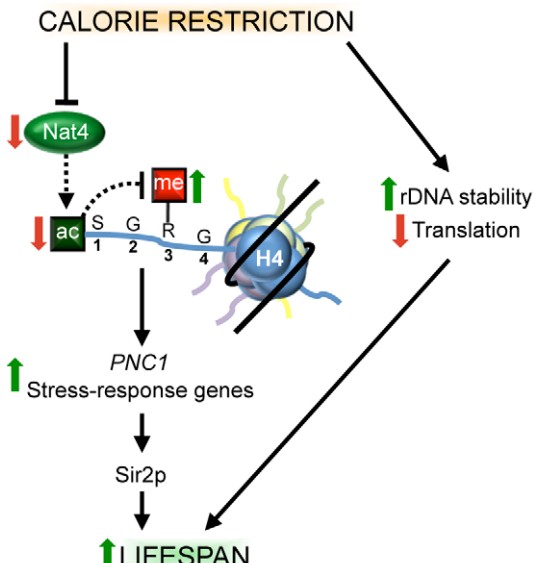

**Figure 7. Model depicting the role of Nat4 and N-acH4 in lifespan.**
*NAT4* expression levels and N-terminal acetylation on histone H4 are reduced under calorie restriction conditions. Lower levels of N-acH4 affect the methylation status at H4R3, with this arginine being a key regulator of the lifespan extension induced by the CR pathway. Under these conditions, several stress-response genes, including *PNC1*, are induced. Lifespan extension is probably an effect of increased Pnc1 activity and the hormesis effect caused by enhanced stress protection previously described. Calorie restriction also extends lifespan through rDNA stability and decrease in translation, pathways that are independent of Nat4.

the proposed model (Fig 7). Moreover, the antagonistic relationship between H4R3me2a and N-acH4 [46] further supports the current results, because lack of arginine methylation in the H4R3K mutant shortens lifespan significantly (Fig 5), while loss of N-acH4 in H4S1D, H4S1A, and *nat4Δ* has the opposite effect on replicative lifespan (Figs 1, 4 and EV3C).

Another histone H4 acetylation mark, namely H4K16ac, has already been implicated in the regulation of yeast lifespan through a mechanism that maintains telomeric silencing [5]. However, we anticipated that N-acH4 regulates lifespan through a distinct mechanism, since we have previously found that deletion of *NAT4* does not impact on telomeric silencing [46]. Therefore, due to the connection of N-acH4 with rRNA expression, we initially hypothesized that *nat4Δ* extends lifespan by altering ribosome stoichiometry and thus reducing mRNA translation, a process that was shown to extend lifespan in various model organisms [33,62–65]. However, polysome profile analysis shows that deletion of *NAT4* did not affect the composition of ribosomes (Fig 2D), indicating that *nat4Δ* induces longevity independently of its role in rDNA silencing. Since lack of Nat4 reduces the abundance of all rRNAs [46], it is likely that the stoichiometry of large and small rRNA subunits is not altered, and therefore, in *nat4Δ*, the ribosomes are still assembled properly and the translational apparatus is unaffected (Fig 2D). The above findings on *nat4Δ* are also consistent with previous reports, which demonstrated that CR and its genetic mimics (i.e. *tor1Δ*) do not mediate their longevity effects through rDNA silencing [21,22].

We then explored the possibility that *nat4Δ* prolongs lifespan through a mechanism that reduces rDNA recombination because of the following reasons. Firstly, we have shown that *nat4Δ* largely mimics CR (Figs 1 and 3) and the latter has been linked to suppression of rDNA recombination in certain cases [26,59,66]. Secondly, *nat4Δ* enhanced RENT binding at the rDNA locus (Fig 2B) and greater RENT binding has been previously connected to increased rDNA stability [23]. Thirdly, the above possibility was supported by the fact that *nat4Δ* induces *PNC1* (Fig 3), whose overexpression has already been correlated to reduced ribosomal DNA amplification through enhanced Sir2 activity [30,59,67] and that *nat4Δ*-mediated longevity is dependent on the presence of Sir2 (Fig 6D). However, despite all the above indications, we did not find a connection between *nat4Δ*-induced longevity and altered rDNA recombination (Fig 2C), suggesting that Nat4 and N-acH4 function within a CR pathway that does not implicate rDNA stability (Fig 7). This suggestion is in line with a recent study which performed a comprehensive screen to show that more than 10% of yeast genes are associated with rDNA stability and *NAT4* was not identified as one of them [68]. In conclusion, the increased binding of RENT to the rDNA locus observed in *nat4Δ* (Fig 2B) will most likely be related to the transcriptional role of Nat4 and N-acH4 at this genomic region [46], since binding of the RENT complex is required for rDNA silencing [51].

But then, how is it possible that *nat4Δ*-associated longevity is mediated through Pnc1 and dependent on Sir2 without involving changes in rDNA recombination? A reasonable explanation is that Pnc1 overexpression by loss of Nat4 still stimulates the activity of Sir2, which subsequently extends lifespan through reported mechanisms that are independent to rDNA recombination and the formation of ERCs [28,69,70]. For instance, Sir2 could affect lifespan by regulating stress resistance, as indicated by the finding that *SIR2* stimulation suppresses the short lifespan phenotype of yeast exposed to hydrogen peroxide [71]. Alternatively, increased activation of Sir2 through Pnc1 could impact on cellular lifespan by suppressing the accumulation of reactive oxygen species [72]. Additional evidence arguing for a Pnc1 longevity effect independently of rDNA recombination comes from data concerning its functional human ortholog, known as nicotinamide phosphoribosyltransferase (Nampt) [73,74]. Nampt is upregulated under glucose limitation and activates SIRT1 [75], but it is well accepted that sirtuin-dependent inhibition of rDNA recombination and ERC formation do not play a role in human cell aging [20]. Thus, we propose a model whereby reduced levels of Nat4 by calorie restriction lead to decreased nucleosomal N-acH4 that subsequently results in increased expression of the longevity factor Pnc1 and promotion of Sir2 activity (Fig 7).

Apart from regulating the expression of Pnc1, Nat4 represses a cohort of other stress-response genes (Fig 3 and Table EV1). Therefore, upon deletion of Nat4 and loss of N-acH4 these genes are induced, leading to a moderate enhancement of cellular stress response. A mild stimulation of stress response could function as a protective mechanism, known as hormesis, which is biologically beneficial during cellular aging [76–79]. Hence, our results support the idea that reduction of N-acH4 induces a modest stress-response state from a young age (Fig 6A) that resembles a hormetic effect, thus preparing cells to counteract severe stress when needed during their lifetime (Fig 7). This idea is also consistent with the suggestion

that CR acts by inducing moderate stress that culminates in enhanced stress resistance and ultimately longevity [80]. Furthermore, a similar CR-mediated longevity pathway, which is distinct from Tor1- and sirtuin-dependent pathways, has been proposed to be initiated by inactivation of the ATP-dependent chromatin remodeling enzyme Isw2. However, this longevity pathway is driven through induction of a different cohort of stress-response genes than those controlled by Nat4. Isw2 inactivation extends replicative lifespan primarily through activation of genes involved in HR-based DNA repair [36]. On the contrary, deletion of Nat4 and loss of N-acH4 activates genes mainly involved in glucose metabolism and energy reserve pathways (Fig 3 and Table EV1). Consistently, previous studies have shown that glucose and nutrient limitations in general reprogram yeast metabolism by promoting the accumulation of energy reserve stores [81,82]. In addition, such cellular metabolic shift occurs during physiological yeast aging [14,83,84], supporting the idea that this stress response constitutes a hormetic effect stimulated by CR or Nat4 deletion. Nat4 was first identified in yeast, and at the time, it was also predicted to have homologues in higher eukaryotes, including mammals [41]. Indeed, it was later shown that its human ortholog hNaa40 has a conserved activity toward histones H4 and H2A [45] and regulates the expression of ribosomal RNAs [47]. Recently, the mouse Nat4 ortholog, also known as Patt1, has been implicated in the aging process because its deficiency in mice attenuates an age-associated liver disease by altering lipid metabolism [48]. Specifically, Patt1 knockout enhances fatty acid oxidation leading to a reduction in fat mass [48]. Interestingly, CR has been associated with increased fatty acid oxidation and decreased fat mass as a result of adiponectin upregulation [85]. This evidence indicates a resemblance of Patt1 deficiency to CR and suggests that mammalian Nat4 orthologs may also function as regulators of lifespan.

All in all, our findings propose that upon depletion of Nat4 and loss of N-acH4, a cellular environment is established that partially mimics a state of calorie restriction. Under this condition, metabolic and stress-response genes become activated, rendering cells more resistant to stress as they age. Therefore, these recent developments provide new leads that will help reveal the mechanisms underlying the anti-aging effect of CR. In particular, future studies could establish whether the model proposed in this report (Fig 7) comprises an epigenetic pathway in human cells controlled by CR that could potentially be exploited in a therapeutic strategy for delaying or reversing age-related diseases.

# Materials and Methods

### Yeast strains

Strains are listed in Table EV4.

### Growth conditions

Cells were grown overnight in liquid or solid rich (YPAD) or minimal (SD) medium at 30°C. Cultures were then diluted to O.D. 0.1 in medium with 2 or 0.1% glucose and grown to O.D. 0.8 within exponential phase. Cells were collected and processed immediately in downstream applications.

### Antibodies

Rabbit polyclonal antibodies were raised against H4R3me2a and N-acH4 by Eurogentec (Belgium). Additional details and characterization of these antibodies are provided in [46,47]. Other antibodies used were as follows: H4 (62-141-13; Millipore), H3 (ab1791; Abcam), Naa40[51] (a gift from Qwei Zhai), and IgG (NB810-56910; Novus Biologicals).

### Replicative lifespan

RLS studies were performed as described previously [86,87]. Exponentially growing cells were placed on a petri dish, and mother and daughter cells were differentiated using a tetrad dissector microscope (SporePlay, Singer SP0-001). Virgin daughter cells were isolated as buds from mother cells and every division was recorded. Data were used to generate a survival growth curve and calculate mean lifespan. The nature of this study is explorative, and thus, the experiments had no pre-specified effect size. A minimum of 60 cells of each wild-type and mutant strain were monitored. WT control cells were always analyzed in parallel to the mutant strains in order to ensure accuracy in our findings. Sample preparation involved randomization, and the analysis was done by two co-authors, who were blinded for the identity of the yeast strains. Statistical analysis was performed by one-way ANOVA using GraphPad Prism data software.

### Gene expression analysis

Total RNA from logarithmically grown (O.D. 0.8) yeast cells was isolated using the hot phenol extraction method [88] and then treated with TURBO DNA-free DNase kit (Ambion, AM1907). Isolated RNA (0.5 μg) was reverse-transcribed with PrimeScript RT reagent kit (TaKaRa, RR820A) using oligo dT primer (50 μM) and random hexamers (100 μM). A negative control reaction was carried out without the RT enzyme. The reaction was diluted by adding 70 μl of DNase RNAse-free water and then analyzed with real-time PCR. Target gene expression levels were measured by SYBR Green-based qPCR (Kapa SYBR Fast Master Mix, KK4602) using gene-specific primers (Table EV5) and a Bio-Rad CFX96 Real-Time PCR system. Expression levels were normalized to the control genes *RPP0, ACT1,* or *TAF10*. Error bars for each sample represent the standard error of the mean. Statistical significance *P*-values were calculated by unpaired two-tailed Student's *t*-test using GraphPad Prism data software.

### ChIP

ChIP assays were performed as described previously [89]. Briefly, the cells were grown to mid-exponential phase at 30°C and then fixed by adding 37% formaldehyde to the media (1.25% final concentration) and incubating with gentle shaking at room temperature for 2 h. Cells were then pelleted and lysed with sonication. The chromatin was immunoprecipitated with the following antibodies: anti-NacH4, anti-H4, and anti-IgG as a negative control. In ChIPs performed with TAP-tagged strains, anti-IgG was used for the immunoprecipitation of the tagged protein, while the untagged equivalent strain was used as a negative control.

A Bio-Rad CFX96 Real-Time PCR system and 96-well plates were used for real-time PCR. Error bars for each sample represent the standard error of the mean. Statistical analysis and calculation of *P*-values were performed by unpaired two-tailed Student's *t*-test using GraphPad Prism data software. The primers used are listed in Table EV5.

### RNA-Seq analysis

Total RNA was extracted as above. Each extraction experiment was performed in independent triplicates. Samples were sequenced at BGI (www.genomics.cn) and EMBL GeneCore facility (www.genecore.embl.de). Paired-end cDNA libraries were prepared from 10 μg of total RNA using the Illumina mRNA-Seq-Sample Prep Kit according to manufacturer's instructions. cDNA fragments of ~400 bp were purified from each library and confirmed for quality by Agilent 2100 Bioanalyzer. High-throughput sequencing was done on an Illumina HiSeq2000 platform, according to the manufacturer's instructions. Quality of the sequenced reads was checked by FastQC 0.11.1 (S. Andrews. FastQC A Quality Control tool for High Throughput Sequence Data. http://www.bioinformatics.babraham.ac.uk/projects/fastqc/). The reads were mapped to the reference yeast genome sequence sacCer3 (UCSC/SGD Apr2011) [90] using GSNAP 2014-09-30 [91]. Used GSNAP settings: gsnap -N 1 -n 1 -O -A sam –merge-distant-samechr -s splicesites sacCer3_Ensemble78.iit. Only uniquely mapped reads were considered for further analysis. Mapped reads were post-processed using SAMtools 0.1.16 [92]. Gene abundance was quantified using HTSeq-count 0.6.1 [93], with the Ensembl gene annotation (release 78) [94] and with taking strandeness of the sequencing protocol into the account. Used HTSeq-count settings: htseq-count -f bam -r pos -s reverse -m union -a 10 -t exon -i gene_id. Gene expression analysis was performed using Bioconductor R package edgeR 3.8.2 [95]. Differential gene expression was calculated using information from two biological replicates for each condition using generalized linear model [96] with tagwise dispersion [97] and FDR correction according to the edgeR vignette. Expression levels were normalized using TMM [98]. Genes were considered as differentially expressed if abs(logFC) ≥ 1 and FDR ≤ 0.0001. Transcriptome data were deposited at the genome expression omnibus repository (GEO) with the accession number GSE86242.

### Gene ontology

Analysis was done using the Database for Annotation Visualization and Integrated Discovery DAVID 6.7 [99]. We used Functional Annotation Clustering to get enriched GO clusters. EASE score [100] was used to select significantly over/under-represented GO clusters with an enrichment score for each cluster given as a mean of −log10p of member GO categories with *P* < 0.05. Only genes expressed in at least one replicate were used as background for the GO clusters calculation.

### Venn diagrams

Venn diagrams were generated using online tool Venny 2.0. Significance of the overlap was evaluated using R package gmp 0.5-12. gmp: Multiple Precision Arithmetic. R package version 0.5-12

(http://CRAN.R-project.org/package = gmp) using hypergeometric *P*-value.

### Western blotting

Western blotting assays were performed as described previously [46]. Briefly, cells were grown to mid-exponential phase at 30°C. Total yeast extracts were prepared by resuspending cell pellets in SDS loading buffer (50 mM Tris–HCl pH 6.8, 2% SDS, 10% glycerol, 1% β-mercaptoethanol, 12.5 mM EDTA, and 0.02% bromophenol blue). Proteins were separated in a 15% SDS–PAGE at 200 V for 1 h, and then, they were wet transferred into a PVDF membrane (GE Healthcare life sciences) with 20% methanol transfer buffer (25 mM Tris, 192 mM glycine, pH 8.3), at 100 V for 1 h. The membrane was blocked in 5% BSA, 0.1% Tween-20 TBS buffer (25 mM Tris, 150 mM NaCl, 2 mM KCl, pH 8) before incubation with the appropriate antibody.

### Polysome profiling analysis

Polysome profiling was performed as described previously [101]. Briefly, strains were grown at 25°C in 100 ml yeast extract–peptone–dextrose medium to early log phase (A600 of 0.6), and subsequently, cycloheximide (final concentration, 100 μg/ml) was added. Extract preparation and sucrose gradient centrifugation were performed as described previously [102].

### Quantification of rDNA copy number and mRNA expression in isolated old and young yeast cells

Purification of replicatively aged yeast cells was performed as described [103]. Genomic DNA was extracted using YeaStar Genomic DNA Kit (Zymo), and relative rDNA copy number was measured by quantitative real-time PCR (Thermo Fisher) using primer sets targeting rDNA loci NTS2-1, RDN58-1, and RDN18-1, as well as control regions (ACT1 and an intergenic region on chromosome V). Measurements from the rDNA loci and the control regions were averaged respectively and compared (*n* = 3). Total RNA was extracted using RNA purification mini kits (QIAGEN), and gene expression was measured by quantitative real-time PCR as described above.

Expanded View for this article is available online.

### Acknowledgements

We thank Danesh Moazed, Thomas Arnersen, Steven Clarke, Markus Proft, and David Sinclair for making their yeast strains available; Patricia Agudelo-Romero, Vasilis Promponas, and Nayia Nicolaou for help with statistical analysis; and Qiwei Zhai for kindly providing the Naa40 antibody. We would also like to acknowledge the contribution of the COST Action CM1406. This work was supported by a grant to AK from the European Research Council (No. 260797, ChromatinModWeb).

### Author contributions

DM-S, VS, ES, DK, GZ, HB, WD, and AK designed the experiments; DM-S and VS performed RNA isolation, reverse-transcription qPCRs, and ChIP assays; VS prepared samples for RNA-sequencing and analyzed data; DM-S and ES constructed mutant yeast strains; DM-S performed immunoblot analysis; ES

and CD performed replicative lifespan experiments; JO analyzed transcriptome data; WL and WD isolated young and old yeast cells, extracted RNA, and analyzed rDNA copy number; GZ and HB performed polysome profiling analysis; and the manuscript was written by AK and critically read by DM-S.

## Conflict of interest

The authors declare that they have no conflict of interest.

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
