## [Review Process File · EMBO Reports]

Manuscript EMBO-2016-42540

Loss of Nat4 and its associated histone H4 N-terminal acetylation mediates calorie restriction-mediated longevity

Diego Molina-Serrano, Vassia Schiza, Christis Demosthenous, Emmanouil Stavrou, Jan Oppelt, Dimitris Kyriakou, Wei Liu, Gertrude Zisser, Helmut Bergler, Weiwei Dang, Antonis Kirmizis

Corresponding author: Antonis Kirmizis, University of Cyprus

Review timeline:

Submission date:	12 April 2016
Editorial Decision:	11 May 2016
Revision received:	08 August 2016
Editorial Decision:	05 September 2016
Revision received:	21 September 2016
Accepted:	30 September 2016

Editor: Esther Schnapp

Transaction Report:

1st Editorial Decision

11 May 2016

Thank you for the submission of your research manuscript to EMBO reports. We have now received the enclosed referee reports on your study that are copied below, as well as referee cross-comments.

As you will see, the referees acknowledge that the findings are potentially interesting. However, they also suggest some more experiments that would be required to strengthen the study. On one hand, more data are needed to confirm that Nat4 regulates lifespan, and on the other hand, it remains somewhat unclear how Pnc1 affects longevity. While strengthening both aspects would be most welcome, establishing a clear role for Nat4 as lifespan regulator downstream of CR is more important for publication of the paper here.

Referee 2 agrees with referee 3 (in the cross-comments) that it should be analyzed whether constitutive Nat4 expression maintains high N-acH4 levels under CR, and whether CR impacts on N-acH4 levels in a Nat4 deletion mutant. Whether constitutive Nat4 expression prevents lifespan extension by CR should also be examined. Both referees further agree that it should be tested how Pcn1 overexpression affects lifespan in nat4delta and sir2/fob1delta cells. These experiments should therefore be performed during revision of the manuscript for EMBO reports.

Referee 2 also indicates in her/his cross-comments that point 2 of referee 1 would not need to be addressed experimentally. The lifespan curves should be repeated and remaining variations explained. Point 5 of referee 1 does also not need to be addressed. Referee 2 further feels that addressing points 3 and 4 of referee 3 would not be strictly required for publication of the paper here.

Given these constructive comments, we would like to invite you to revise your manuscript with the understanding that the referee concerns must be addressed and their suggestions taken on board. Please address all referee concerns in a complete point-by-point response. Acceptance of the manuscript will depend on a positive outcome of a second round of review. It is EMBO reports policy to allow a single round of revision only and acceptance or rejection of the manuscript will therefore depend on the completeness of your responses included in the next, final version of the manuscript.

Revised manuscripts should be submitted within three months of a request for revision; they will otherwise be treated as new submissions. Please contact us if a 3-months time frame is not sufficient for the revisions so that we can discuss this further. I suggest that you layout the manuscript as a normal article for which there are no length limitations. Please note that supplementary figures and tables are called expanded view (EV) now. These data are integrated into the manuscript text online and expand when clicked. Please upload EV figures and tables as separate files and add the figure legends to the end of the main manuscript file.

Regarding data quantification, please specify the number "n" for how many experiments were performed, the bars and error bars (e.g. SEM, SD) and the test used to calculate p-values in the respective figure legends. This information must be provided in the figure legends. Please also include scale bars in all microscopy images.

We now strongly encourage the publication of original source data with the aim of making primary data more accessible and transparent to the reader. The source data will be published in a separate source data file online along with the accepted manuscript and will be linked to the relevant figure. If you would like to use this opportunity, please submit the source data (for example scans of entire gels or blots, data points of graphs in an excel sheet, additional images, etc.) of your key experiments together with the revised manuscript. Please include size markers for scans of entire gels, label the scans with figure and panel number, and send one PDF file per figure or per figure panel.

- a complete author checklist, which you can download from our author guidelines (<http://embor.embopress.org/authorguide#revision>). Please insert page numbers in the checklist to indicate where in the manuscript the requested information can be found.
- a letter detailing your responses to the referee comments in Word format (.doc)
- a Microsoft Word file (.doc) of the revised manuscript text
- editable TIFF or EPS-formatted figure files in high resolution
- a separate PDF file of any Supplementary information (in its final format)

As part of the EMBO publication's Transparent Editorial Process, EMBO reports publishes online a Review Process File to accompany accepted manuscripts. This File will be published in conjunction with your paper and will include the referee reports, your point-by-point response and all pertinent correspondence relating to the manuscript.

I look forward to seeing a revised version of your manuscript when it is ready. Please let me know if you have questions or comments regarding the revision.

REFeree REPORTS

Referee #1:

Review of Molina-Serrano et al.

In this study, Molina-Serrano et al. report that deletion of Nat4 increases replicative lifespan in yeast through a mechanism involving altered N-terminal acetylation and activation of Pnc1. Overall, this study has the potential to be of high interest to yeast the aging field. There are a couple of experimental issues that should be addressed, and I found the presentation of the story to detract somewhat from the overall impact.

Major Comments:

1. In my mind, the structure of the manuscript detracted from the impact. The authors spend the first couple of figures looking at the rDNA and mRNA translation, which turns out not to be important for the overall story at all, aside from ruling out a potential mechanism, as far as I can tell. It seems like the model the authors want to argue for is that the gene expression changes associated with loss of Nat4 through chromatin modifications result in enhanced stress response and lifespan extension. A secondary point is that this mimics to some extent what caloric restriction does in yeast. These should be the focus of the manuscript, not rDNA and translation. I realize this is stylistic, so it's more of a suggestion, but I think the impact would be enhanced by reorganizing along these lines.

2. The Pnc1 connection is, in my opinion, unconvincing. Without some idea for how Pnc1 might be mediating the downstream stress response, it doesn't make much sense. Is this thought to be through Sir2? But then it would be rDNA... Are the nat4 cells actually stress resistant and is that dependent on Pnc1? Are the relevant gene expression changes dependent on Pnc1? The lifespan data is also problematic in that the pnc1 single mutant clearly has a different shaped survival curve relative to the other strains, perhaps implying that this strain is dying for other reasons, so failure to show as large of a lifespan extension in this context is difficult to interpret. The relative effect seems to be amplified by the exceptionally long (compared to other figure panels) lifespan of the nat4 single mutant in this experiment. Related to this, the pnc1 nat4 double mutant lifespan in this experiment is comparable to the nat1 lifespan in other figure panels. Since this is a major component of the proposed model, I think it's important for the authors to clean this up so that readers can have confidence in the reported results.

3. In general, the lifespan curves are shorter than what is typical in the yeast aging literature. Do the authors have an explanation for this? I realize that different conditions can influence baseline phenotypes, but it lessens confidence in the results when the controls are not in line with a majority of the literature in the field.

Minor comments:

4. It's not clear from the methods how much glucose is actually present in the cultures at the times that the assays for gene expression are performed. How much glucose is used up in outgrowth to "mid-log phase" under these conditions? This is important both for interpretation of the meaning of the data, but also so that others will be able to replicate these findings.

5. Are a subset of changes in gene expression mediated by Msn2/Msn4? This could tie together the CR, tor1, and nat4 connection outside of the rDNA

Referee #2:

Reviewer comments:

Overall this is a very nice and clear manuscript and a further demonstration of the links between environmental signals that are incorporated as histone modifications to affect cellular aging. The finding that nat4 Δ is epistatic to calorie restriction-induced longevity is exciting. Given the quality and scope of the manuscript, if the detailed comments below can be addressed, the paper is suitable for publication in EMBO Reports.

1. The authors claim in Figure 1A that NAT4 expression decreases with calorie restriction. While the NAT4 RNA levels have indeed been shown to decrease, it would be more convincing if the authors could show that the level of NAT4 protein is also decreasing with CR, e.g. by western blot.
2. Figure 1E is missing nat4Δ NCR.
3. In Figure 2A please explain abbreviations (NTS, ETS, ITS etc) either directly in the figure or in the figure legend.
4. On page 6 of the manuscript and in Figure 2D, the authors claim that ribosome biogenesis is not affected in nat4Δ. However, in the RNA-seq data in the subsequent results section on page 7 and Figure 3A ribonucleoprotein biogenesis is shown to be downregulated. Though this contradiction is addressed in the discussion on page 12, it is confusing when reading the results section. Some extra clarifications in the results section would be helpful.
5. While the experiments in the manuscript have shown the effect of deleting Nat4 and loss of N-acH4 on lifespan, it would be interesting to know the effects of overexpressing Nat4. Specific questions that could be addressed are:
What is the effect of overexpressing Nat4 on N-acH4 level in the rDNA region?
How does overexpression of Nat4 impact lifespan?

Referee #3:

In this paper, Molina-Serrano et al., link H4 N-terminal acetylation by Nat4 to replicative lifespan control by caloric restriction (CR) in the yeast *S. cerevisiae*. They conclude that down-regulation of Nat4 by CR is extending lifespan by elevating the levels of the "stress response" protein Pnc1, a nicotinamidase that is known to affect Sir2 activity by virtue of consuming nicotinamide; a negative regulator of Sir2. However, based on data presented it is argued that Nat4 is regulating lifespan in a Sir2-independent, but Pcn1-dependent manner.

This is an interesting and well-written paper. At present, however, there are a few more experiments needed to back up their model and the conclusions made as outlined below:

1. The data presented in figure 1 and the conclusion made upon these experiments, i.e. that CR is reducing N-acH4 by down-regulating the levels of Nat4 is certainly possible but a causative link is not shown. To firmly establish that CR is indeed acting through Nat4 here one would need to test if N-acH4 levels fail to be reduced by CR when Nat4 is constitutively expressed by a promoter that are not regulated by CR (for example by the GPD promoter). Similarly, one could test if CR fails, or not, to further down-regulate N-acH4 levels in a nat4 deletion mutant. An alternative would be to be a bit more circumspect in the conclusions here.
2. Additionally, to firmly establish that CR acts through Nat4 in extending lifespan one would need to check if CR fails to extend lifespan when Nat4 is constitutively expressed. The reason for the nat4 deletion mutant not responding to CR could be that the "roof" of lifespan extension has already been reached. Also, does a nat4Δ deletion extend lifespan in a pde2Δ mutant; i.e. a mutant that do respond to CR.
3. In relation to a possible link between N-acH4 levels, Nat4, and lifespan, the lifespan of a nat4Δ and H4S1D double mutant should be tested (Fig. 4B).
4. The conclusion that Nat4 is not affecting lifespan through effects on 60S ribosome subunits should be supplemented by testing the effect of a nat4 deletion on the lifespan of a gen4Δ mutant (Fig. 2).
5. Since the effects of Sir2 (and Pnc1) has been argued by some laboratories to be mostly independent on ERC levels and rather linked to the regulation of recombination activity at the rDNA locus, the model (Figure 7) needs some more experimental backup to establish that the rDNA stability route and the Nat4/Pnc1 route are indeed parallel. The model explicitly predicts that

overproduction of Pnc1 should prolong lifespan on its own since it is acting downstream of Nat4. Thus, the effect of Pnc1 overproduction on the lifespan of *nat4Δ* cells and *sir2Δfob1Δ* cells should be tested.

Minor points

6. Not sure references 34&35 show that CR enhances stress response pathways through inhibition of ATP-dependent chromatin remodeling - the sentence on p. 3 appear to indicate this.

1st Revision - authors' response

08 August 2016

Response to Comments

Thank you for the submission of your research manuscript to EMBO reports. We have now received the enclosed referee reports on your study that are copied below, as well as referee cross-comments.

As you will see, the referees acknowledge that the findings are potentially interesting. However, they also suggest some more experiments that would be required to strengthen the study. On one hand, more data are needed to confirm that Nat4 regulates lifespan, and on the other hand, it remains somewhat unclear how Pnc1 affects longevity. While strengthening both aspects would be most welcome, establishing a clear role for Nat4 as lifespan regulator downstream of CR is more important for publication of the paper here.

Referee 2 agrees with referee 3 (in the cross-comments) that it should be analyzed whether constitutive Nat4 expression maintains high N-acH4 levels under CR, and whether CR impacts on N-acH4 levels in a Nat4 deletion mutant. Whether constitutive Nat4 expression prevents lifespan extension by CR should also be examined. Both referees further agree that it should be tested how Pnc1 overexpression affects lifespan in *nat4Δ* and *sir2/fob1Δ* cells. These experiments should therefore be performed during revision of the manuscript for EMBO reports.

To address the points that have been raised by referees we have constructed a new yeast strain in which the expression of Nat4 is driven by the constitutive STE5 promoter (*Pste5-NAT4*). This promoter drives transcription of Nat4 at similar levels to the native *NAT4* promoter and, importantly, its activity is not affected by CR (see revised Fig 1A). Using this newly constructed yeast strain (*Pste5-NAT4*), we show in new figures 1B and 6E that constitutive expression of Nat4 prevents the reduction of N-acH4 levels under CR compared to a WT strain (compare green and blue bars). Furthermore, these figures show that CR does not further impact on the N-acH4 levels in *nat4Δ* mutant compared to WT cells (Figs 1B and 6E, compare red and blue bars in CR) suggesting that CR impacts on this histone modification mainly through Nat4.

In figure 1F of the revised manuscript we show that constitutive expression of Nat4 in the *Pste5-NAT4* strain limits the extension of lifespan mediated by CR compared to the longevity observed in a WT strain (from 32% down to 19%). This result supports the conclusion of this study presented in figure 7, which demonstrates that Nat4 functions within one of the pathways induced by CR to regulate cellular lifespan.

While we agree with the referees' point that the connection between Pnc1 and *nat4Δ*-induced longevity needs further exploration, we believe that overexpressing Pnc1 in *nat4Δ* cells would not provide further information on this issue because Pnc1 is already overexpressed in *nat4Δ* mutants (Figs 3C, 4C, 4D, 6A and Table EV1). Alternatively, we examined lifespan in *nat4Δ* cells lacking the transcription factors Msn2 and Msn4 which directly activate the expression of Pnc1 (Medvedik et al, PLoS Biology 2007) and we show that the triple mutant *msn2Δ msn4Δ nat4Δ* is no longer able to extend lifespan (Fig 6C). This is another evidence reinforcing the idea that Pnc1 induction is required for the extension of lifespan in *nat4Δ* cells.

Finally, we examined lifespan in *sir2Δfob1Δ* cells overexpressing Pnc1 and found that lifespan is not extended indicating that Pnc1-mediated longevity is dependent on Sir2 as previously described (Anderson et al., Nature 2003). We also examined lifespan in *sir2Δfob1Δ* cells lacking Nat4 and found that *nat4Δ* cannot extend lifespan in this mutant background, resembling the effect of Pnc1 above. Therefore, this further verifies that Nat4 and Pnc1 function within the same pathway. These additional data are described in the results section and shown in figures 6D and EV6.

Referee 2 also indicates in her/his cross-comments that point 2 of referee 1 would not need to be addressed experimentally. The lifespan curves should be repeated and remaining variations explained. Point 5 of referee 1 does also not need to be addressed. Referee 2 further feels that addressing points 3 and 4 of referee 3 would not be strictly required for publication of the paper here.

Lifespan curves have been obtained after analyzing a minimum of 60 independent cells for each mutant, and examining in parallel WT control cells during each RLS experiment. The mean lifespan numbers obtained for each of the various yeast genetic background strains used have been consistently reproducible during the implementation of this study. We performed additional lifespan experiments for *pnc1*Δ strains as recommended and we now show in revised figure 6B that all strains tested have similarly shaped survival curves. As in the previous version of the manuscript, these results show that loss of Pnc1 prevents *nat4*Δ-mediated longevity.

Given these constructive comments, we would like to invite you to revise your manuscript with the understanding that the referee concerns must be addressed and their suggestions taken on board. Please address all referee concerns in a complete point-by-point response. Acceptance of the manuscript will depend on a positive outcome of a second round of review. It is EMBO reports policy to allow a single round of revision only and acceptance or rejection of the manuscript will therefore depend on the completeness of your responses included in the next, final version of the manuscript.

Revised manuscripts should be submitted within three months of a request for revision; they will otherwise be treated as new submissions. Please contact us if a 3-months time frame is not sufficient for the revisions so that we can discuss this further. I suggest that you layout the manuscript as a normal article for which there are no length limitations. Please note that supplementary figures and tables are called expanded view (EV) now. These data are integrated into the manuscript text online and expand when clicked. Please upload EV figures and tables as separate files and add the figure legends to the end of the main manuscript file.

Regarding data quantification, please specify the number "n" for how many experiments were performed, the bars and error bars (e.g. SEM, SD) and the test used to calculate p-values in the respective figure legends. This information must be provided in the figure legends. Please also include scale bars in all microscopy images.

We now strongly encourage the publication of original source data with the aim of making primary data more accessible and transparent to the reader. The source data will be published in a separate source data file online along with the accepted manuscript and will be linked to the relevant figure. If you would like to use this opportunity, please submit the source data (for example scans of entire gels or blots, data points of graphs in an excel sheet, additional images, etc.) of your key experiments together with the revised manuscript. Please include size markers for scans of entire gels, label the scans with figure and panel number, and send one PDF file per figure or per figure panel.

- a complete author checklist, which you can download from our author guidelines (<http://embor.embopress.org/authorguide#revision>). Please insert page numbers in the checklist to indicate where in the manuscript the requested information can be found.
- a letter detailing your responses to the referee comments in Word format (.doc)
- a Microsoft Word file (.doc) of the revised manuscript text
- editable TIFF or EPS-formatted figure files in high resolution
- a separate PDF file of any Supplementary information (in its final format)

As part of the EMBO publication's Transparent Editorial Process, EMBO reports publishes online a Review Process File to accompany accepted manuscripts. This File will be published in conjunction with your paper and will include the referee reports, your point-by-point response and all pertinent correspondence relating to the manuscript.

I look forward to seeing a revised version of your manuscript when it is ready. Please let me know if you have questions or comments regarding the revision.

Referee #1:

Review of Molina-Serrano et al.

In this study, Molina-Serrano et al. report that deletion of Nat4 increases replicative lifespan in yeast through a mechanism involving altered N-terminal acetylation and activation of Pnc1. Overall, this study has the potential to be of high interest to yeast the aging field. There are a couple of experimental issues that should be addressed, and I found the presentation of the story to detract somewhat from the overall impact.

Major Comments:

1. In my mind, the structure of the manuscript detracted from the impact. The authors spend the first couple of figures looking at the rDNA and mRNA translation, which turns out not to be important for the overall story at all, aside from ruling out a potential mechanism, as far as I can tell. It seems like the model the authors want to argue for is that the gene expression changes associated with loss of Nat4 through chromatin modifications result in enhanced stress response and lifespan extension. A secondary point is that this mimics to some extent what caloric restriction does in yeast. These should be the focus of the manuscript, not rDNA and translation. I realize this is stylistic, so it's more of a suggestion, but I think the impact would be enhanced by reorganizing along these lines.

We appreciate the Referee's suggestion regarding the structure of the manuscript. However, after thorough discussion among the authors we decided to keep the structure of the paper as initially presented because it reflects better the development of the overall story according to previously known facts about Nat4. Specifically, at the beginning of this study it was known, according to our previous findings (Schiza et al., PLoS Genetics, 2013), that Nat4 and its associated modification N-acH4 regulate rDNA silencing. Therefore, considering that rDNA-associated processes play an important role in the control of cellular aging we believe it is necessary to emphasize that Nat4 and N-acH4 are not implicated in cellular lifespan through their previous connection to rDNA.

2. The Pnc1 connection is, in my opinion, unconvincing. Without some idea for how Pnc1 might be mediating the downstream stress response, it doesn't make much sense. Is this thought to be through Sir2? But then it would be rDNA... Are the nat4 cells actually stress resistant and is that dependent on Pnc1? Are the relevant gene expression changes dependent on Pnc1? The lifespan data is also problematic in that the pnc1 single mutant clearly has a different shaped survival curve relative to the other strains, perhaps implying that this strain is dying for other reasons, so failure to show as large of a lifespan extension in this context is difficult to interpret. The relative effect seems to be amplified by the exceptionally long (compared to other figure panels) lifespan of the nat4 single mutant in this experiment. Related to this, the pnc1 nat4 double mutant lifespan in this experiment is comparable to the nat1 lifespan in other figure panels. Since this is a major component of the proposed model, I think it's important for the authors to clean this up so that readers can have confidence in the reported results.

In order to strengthen the Pnc1 connection we performed lifespan experiments in *nat4Δ* cells lacking the transcription factors Msn2 and Msn4 which directly activate the expression of *PNC1* (Medvedik et al, PLoS Biology 2007). We now show that the triple mutant *msn2Δ msn4Δ nat4Δ* is unable to extend lifespan (Fig 6C). Furthermore, we examined lifespan in *sir2Δ fob1Δ* cells overexpressing Pnc1 and found that lifespan is not extended indicating that Pnc1-mediated longevity is dependent on Sir2 as previously described (Anderson et al., Nature 2003). We also examined lifespan in *sir2Δ fob1Δ* cells lacking Nat4 and found that *nat4Δ* cannot extend lifespan in this

mutant background, resembling the effect of Pnc1 above. Therefore, these new data which are presented in figures 6C, 6D and EV6 provide further evidence that Nat4 and Pnc1 function within the same longevity pathway. These results also support the idea that Nat4 and Pnc1 regulate lifespan via Sir2 in an rDNA-independent manner, through other reported Sir2-dependent mechanisms. The fact that Nat4 does not function through the rDNA stability route is also supported by a recent paper which identified through a comprehensive screen more than 10% of yeast genes to be related with rDNA maintenance but Nat4 was not one of them (Saka K et al., Nucleic Acids Res 2016). We elaborate on this conclusion within the discussion section. Moreover, we believe that determining further how Pnc1 mediates the downstream stress response would be a major advancement in itself and therefore, beyond the scope of this paper.

To address the referees comment on the *pnc1Δ* lifespan curve, we performed additional RLS experiments for all the strains indicated in figure 6B. We obtained similarly shaped survival curves for all strains and still observe that the double deletion *pnc1Δ nat4Δ* does not prolog lifespan to the same extent as the *nat4Δ* single mutant (revised Fig 6B), supporting the idea that Pnc1 is required for *nat4Δ*-induced longevity. Indeed, the lifespan of the *nat4Δ* single mutant generated within these strains is slightly longer (lifespan extension of 30%) than that observed in other RLS experiments. However, in general we observe lifespan extension upon Nat4 deletion in different strain backgrounds ranging from 21% to 41% as indicated in figure 1D and this observation is described within the results section. This range in lifespan extension is probably due to genetic variations between mutant strains generated for the various RLS experiments. To control for these variations and have confidence on the obtained results, we always examine the lifespan of WT cells in parallel to the mutant strains within each RLS experiment.

3. In general, the lifespan curves are shorter than what is typical in the yeast aging literature. Do the authors have an explanation for this? I realize that different conditions can influence baseline phenotypes, but it lessens confidence in the results when the controls are not in line with a majority of the literature in the field.

We thank the referee for pointing out this issue. This is something we have also realized while carrying out this work and that is why in every RLS experiment we analyze a minimum of 60 cells and include WT control cells in parallel to the mutant strains in order to ensure accuracy in our findings. We clarify this within Materials & Methods. Furthermore, it is also worth noting that other important studies used the same yeast background strains and reported yeast lifespan numbers and curves (Anderson et al., Nature 2003; Dang et al., Cell Metabolism 2014 [Figures 2]; Mei and Brenner, PLoS Biology 2015, etc.) similar to the ones described in this manuscript. In agreement with the referee's comment, we believe that the baseline differences in RLS experiments among reported studies are most likely attributed to differences in cell culture conditions (i.e. media, incubation times, preculture methods) and variations in the genetic background of yeast strains used within various laboratories.

Minor comments:

4. It's not clear from the methods how much glucose is actually present in the cultures at the times that the assays for gene expression are performed. How much glucose is used up in outgrowth to "mid-log phase" under these conditions? This is important both for interpretation of the meaning of the data, but also so that others will be able to replicate these findings.

We note in Materials & Methods that yeast cultures for every experiment were initiated at dilution O.D. 0.1 with media containing the indicated glucose concentration (2% or 0.1%) and then cells were harvested during the exponential growth phase at O.D. 0.8. At this O.D. value cells are still in growth phase and therefore have not exhausted glucose from the media.

5. Are a subset of changes in gene expression mediated by Msn2/Msn4? This could tie together the CR, tor1, and *nat4* connection outside of the rDNA

We thank the referee for this insightful comment. To explore the connection between Msn2/Msn4 with the *nat4Δ* phenotype we generated a triple mutant *msn2Δ msn4Δ nat4Δ* and tested its lifespan in RLS assays. We found that lifespan extension is abolished in the triple mutant (presented in Fig 6C), a result that consents with the fact that *nat4Δ*-mediated lifespan extension requires Pnc1 activation.

Referee #2:

Reviewer comments:

Overall this is a very nice and clear manuscript and a further demonstration of the links between environmental signals that are incorporated as histone modifications to affect cellular aging. The finding that *nat4Δ* is epistatic to calorie restriction-induced longevity is exciting. Given the quality and scope of the manuscript, if the detailed comments below can be addressed, the paper is suitable for publication in EMBO Reports.

1. The authors claim in Figure 1A that NAT4 expression decreases with calorie restriction. While the NAT4 RNA levels have indeed been shown to decrease, it would be more convincing if the authors could show that the level of NAT4 protein is also decreasing with CR, e.g. by western blot.

We have attempted the experiment suggested by the referee but the available antibodies, which have been raised against the human ortholog (Naa40), cannot detect yeast Nat4 (Fig EV5). Therefore, we have also HA-tagged the endogenous Nat4 but again we could not detect the protein with the HA antibody (data not shown) because Nat4 is expressed at very low levels. Despite the inability to detect Nat4 protein, we believe the finding showing that Nat4 RNA levels decrease under CR (Fig 1A and Table EV3) is supplemented by the reduction of the Nat4-catalysed modification N-acH4 in CR (Fig 1B and 6E).

2. Figure 1E is missing *nat4Δ* NCR.

We have now performed additional RLS experiments for all strains in this figure and have included the data on *nat4Δ* NCR in revised figure 1E.

3. In Figure 2A please explain abbreviations (NTS, ETS, ITS etc) either directly in the figure or in the figure legend.

We thank the referee for spotting this. We have now explained these abbreviations in the figure legend.

4. On page 6 of the manuscript and in Figure 2D, the authors claim that ribosome biogenesis is not affected in *nat4Δ*. However, in the RNA-seq data in the subsequent results section on page 7 and Figure 3A ribonucleoprotein biogenesis is shown to be downregulated. Though this contradiction is addressed in the discussion on page 12, it is confusing when reading the results section. Some extra clarifications in the results section would be helpful.

Following the referee's suggestion we clarified this contradiction in the results section. Additionally, in the title of the section referring to figure 2D we replaced the words "ribosome biogenesis" with "polysome profiling" for further clarification.

5. While the experiments in the manuscript have shown the effect of deleting Nat4 and loss of N-acH4 on lifespan, it would be interesting to know the effects of overexpressing Nat4. Specific questions that could be addressed are:

What is the effect of overexpressing Nat4 on N-acH4 level in the rDNA region?

How does overexpression of Nat4 impact lifespan?

We have generated a new strain (*Pste5-NAT4*) in which Nat4 is constitutively expressed at physiological levels even under CR conditions (revised Fig 1A). Using this strain we now show that constitutive Nat4 expression maintains high N-acH4 levels under CR (Figs 1B and 6E), reduces the induction of stress response genes in CR (Fig 3D), and limits lifespan extension (from 32% down to 19%) mediated by CR (Fig 1F).

Referee #3:

In this paper, Molina-Serrano et al., link H4 N-terminal acetylation by Nat4 to replicative lifespan

control by caloric restriction (CR) in the yeast *S. cerevisiae*. They conclude that down-regulation of Nat4 by CR is extending lifespan by elevating the levels of the "stress response" protein Pnc1, a nicotinamidase that is known to affect Sir2 activity by virtue of consuming nicotinamide; a negative regulator of Sir2. However, based on data presented it is argued that Nat4 is regulating lifespan in a Sir2-independent, but Pnc1-dependent manner.

This is an interesting and well-written paper. At present, however, there are a few more experiments needed to back up their model and the conclusions made as outlined below:

1. The data presented in figure 1 and the conclusion made upon these experiments, i.e. that CR is reducing N-acH4 by down-regulating the levels of Nat4 is certainly possible but a causative link is not shown. To firmly establish that CR is indeed acting through Nat4 here one would need to test if N-acH4 levels fail to be reduced by CR when Nat4 is constitutively expressed by a promoter that are not regulated by CR (for example by the GPD promoter). Similarly, one could test if CR fails, or not, to further down-regulate N-acH4 levels in a *nat4* deletion mutant. An alternative would be to be a bit more circumspect in the conclusions here.

Following the referee's suggestion, we have generated a new strain (*Pste5-NAT4*) in which constitutive Nat4 expression is driven by the *STE5* promoter whose activity is similar to that of the native *NAT4* promoter and is insensitive to CR (Fig 1A). Using this strain we now show that constitutive Nat4 expression maintains high N-acH4 levels under CR (Figs 1B and 6E), reduces the induction of stress response genes in CR (Fig 3D), and limits lifespan extension (from 32% down to 19%) mediated by CR (Fig 1F). Furthermore, we show that CR does not further reduce N-acH4 levels in a *nat4*Δ mutant (Fig 1B and 6E), indicating that Nat4 is the main enzyme through which CR controls N-acH4.

2. Additionally, to firmly establish that CR acts through Nat4 in extending lifespan one would need to check if CR fails to extend lifespan when Nat4 is constitutively expressed. The reason for the *nat4* deletion mutant not responding to CR could be that the "roof" of lifespan extension has already been reached. Also, does a *nat4*Δ deletion extend lifespan in a *pde2*Δ mutant; i.e. a mutant that do respond to CR.

As described in the previous comment, we found that constitutive expression of Nat4 reduces the extension of lifespan mediated by CR from 32% to 19% (Fig 1F). We believe that constitutive Nat4 expression does not completely block the longevity effect of CR because various parallel pathways are induced during CR and some do not involve Nat4 (Fig 7).

3. In relation to a possible link between N-acH4 levels, Nat4, and lifespan, the lifespan of a *nat4*Δ and H4S1D double mutant should be tested (Fig. 4B).

We decided not to proceed with this experiment because based on our previous findings (Schiza et al, PLoS Genetics 2013) Nat4 mediates its effects entirely through N-acH4 and is the main enzyme catalyzing this histone modification. Hence, the levels of N-acH4 cannot be further reduced in this double mutant. Alternatively, to further verify the connection between N-acH4 and lifespan we performed RLS and gene expression analysis in a second histone point mutant (H4S1A) that cannot be N-terminally acetylated by Nat4. We found that, similarly to H4S1D, the H4S1A mutant extends replicative lifespan and induces stress-response genes (Fig EV4).

4. The conclusion that Nat4 is not affecting lifespan through effects on 60S ribosome subunits should be supplemented by testing the effect of a *nat4* deletion on the lifespan of a *gcn4*Δ mutant (Fig. 2).

We believe that supplementing the polysome profiling analysis is no longer necessary because we have generated and included in the revised manuscript several additional data which support the link between Nat4 and a CR-associated pathway that involves Pnc1 activation.

5. Since the effects of Sir2 (and Pnc1) has been argued by some laboratories to be mostly independent on ERC levels and rather linked to the regulation of recombination activity at the rDNA locus, the model (Figure 7) needs some more experimental backup to establish that the rDNA stability route and the Nat4/Pnc1 route are indeed parallel. The model explicitly predicts that

overproduction of Pnc1 should prolong lifespan on its own since it is acting downstream of Nat4. Thus, the effect of Pnc1 overproduction on the lifespan of *nat4Δ* cells and *sir2Δfob1Δ* cells should be tested.

Following the referee's recommendation we have examined lifespan in *sir2Δ fob1Δ* cells overexpressing Pnc1 and found that lifespan is not extended (Fig EV6) indicating that Pnc1-mediated longevity is dependent on Sir2, as previously described (Anderson et al., Nature 2003). In addition, we examined lifespan in *sir2Δ fob1Δ* cells lacking Nat4 and found that *nat4Δ* cannot extend lifespan in this mutant background (Fig 6D), resembling the effect of Pnc1 above. Therefore, this new evidence further verifies that Nat4 and Pnc1 function within the same pathway, which is dependent on Sir2. These data suggest that Nat4 and Pnc1 regulate lifespan via Sir2 in an rDNA-independent manner through other reported Sir2-dependent mechanisms. The fact that Nat4 does not function through the rDNA stability route is also supported by a recent paper which identified through a comprehensive screen more than 10% of yeast genes to be related with rDNA maintenance but Nat4 was not one of them (Saka K et al., Nucleic Acids Res 2016). The implications of the newly generated data are described in the results and discussion sections.

We believe that overexpressing Pnc1 in *nat4Δ* cells would not provide further information on the proposed model because Pnc1 is already overexpressed in *nat4Δ* mutants (Figs 3C, 4C, 4D, 6A and Table EV1). Alternatively, we examined lifespan in *nat4Δ* cells lacking the transcription factors Msn2 and Msn4 which directly activate the expression of Pnc1 (Medvedik et al, PLoS Biology 2007) and we show that the triple mutant *msn2Δ msn4Δ nat4Δ* is no longer able to extend lifespan (Fig 6C). This is another evidence connecting Pnc1 induction with *nat4Δ*-mediated longevity.

Minor points

6. Not sure references 34&35 show that CR enhances stress response pathways through inhibition of ATP-dependent chromatin remodeling - the sentence on p. 3 appear to indicate this.

We thank the referee for spotting this. We re-phrased the sentence so that we are not associating those specific references with inhibition of ATP-dependent chromatin remodeling.

2nd Editorial Decision

05 September 2016

Thank you for the submission of your revised manuscript to our journal. We have now received the comments from referees 1 and 2; referee 3 was unfortunately not available to assess the revised manuscript for us. I have asked referee 2 to please also examine how well referee 3's concerns were addressed.

As you will see, while the referees acknowledge that the study has been strengthened and improved, both also still have remaining concerns that need to be taken into account. These are the major 4 points/suggestions:

- Pnc1 overexpression in the BY4741 background
- show that the cells grown in 0.1% glucose to OD 0.8 are still in log/growth phase
- demonstrate statistical significance in the new figures (this needs to be done for all figures!)
- show that Pnc1 is not induced in the *msn2Δ msn4Δ nat4Δ* strain in comparison to *nat4Δ* cells

I think that all 4 points should be addressed experimentally. If point 1 cannot be done, please discuss the discrepancy with earlier studies and provide potential explanations. I would therefore like to give you the opportunity to address the remaining concerns in a second revision that should be performed and the final manuscript submitted as soon as possible.

The manuscript currently has 7 main plus 7 EV figures. Unfortunately, at the moment, we can only offer 5 EV figures per manuscript. However, your EV figures are predominantly single panels only, and could potentially be combined into a total of 5 EV figures.

EMBO press papers are accompanied online by A) a short (1-2 sentences) summary of the findings and their significance, B) 2-3 bullet points highlighting key results and C) a synopsis image that is

550x200-400 pixels large (the height is variable). You can either show a model or key data in the synopsis image. Please note that text needs to be readable at the final size. Please send us this information along with the revised manuscript.

REFEREE REPORTS

Referee #1:

The authors have added some important experimental evidence in response to the major referee comments. I am still concerned, however, that the manuscript is structured awkwardly and fails to make a convincing and clear case for the model that the authors propose. In particular, the sir2 fob1 data while an important addition is also difficult to interpret, since the authors argue that nat4 deletion mimics CR, yet several groups have published that CR increases lifespan more robustly in the sir2 fob1 double mutant compared to WT, as does deletion of tor1. Does CR still extend lifespan of the sir2 fob1 nat4 triple mutant? Either Nat4 is in the same pathway with CR and TOR or it isn't. The msn2 msn4 data are also not consistent with prior results where this double deletion mutant is long-lived. The PSY316 strain should be avoided for these studies, as activation of Sir2 fails to extend lifespan in that strain, a major flaw with some of the prior literature on Pnc1.

I also do not accept the authors' explanation in the rebuttal to my question about the glucose concentration in the culture medium. The fact that the cells were harvested at an OD of 0.8, which the authors claim is still log-phase for cells initiated in 0.1% glucose, suggests a possible lack of experimental rigor. I find it hard to believe that cells started in a 0.1% glucose culture will not have exhausted nearly all of the glucose by the time they reach OD of 0.8, while cells started in 2% glucose will certainly have exhausted a significant portion of the glucose. This can and should be quantified and experimentally corrected if necessary.

Referee #2:

Response to Author rebuttal

Briefly, the authors have nicely addressed the concerns and requests of the reviewers. For publication in EMBO reports, I would suggest that the authors include statistical significance in their new figures as well as showing that Pnc1 is not induced in the msn2 Δ msn4 Δ nat4 Δ strain in comparison to nat4 Δ cells by RT-qPCR for example. Please see the point-by-point responses below.

Dear Antonis,

Thank you for the submission of your research manuscript to EMBO reports. We have now received the enclosed referee reports on your study that are copied below, as well as referee cross-comments. As you will see, the referees acknowledge that the findings are potentially interesting. However, they also suggest some more experiments that would be required to strengthen the study. On one hand, more data are needed to confirm that Nat4 regulates lifespan, and on the other hand, it remains somewhat unclear how Pnc1 affects longevity. While strengthening both aspects would be most welcome, establishing a clear role for Nat4 as lifespan regulator downstream of CR is more important for publication of the paper here. Referee 2 agrees with referee 3 (in the cross-comments) that it should be analyzed whether constitutive Nat4 expression maintains high N-acH4 levels under CR, and whether CR impacts on N-acH4 levels in a Nat4 deletion mutant. Whether constitutive Nat4 expression prevents lifespan extension by CR should also be examined. Both referees further agree that it should be tested how Pnc1 overexpression affects lifespan in nat4 Δ and sir2/fob1 Δ cells. These experiments should therefore be performed during revision of the manuscript for EMBO reports.

To address the points that have been raised by referees we have constructed a new yeast strain in which the expression of Nat4 is driven by the constitutive STE5 promoter (Pste5-NAT4). This promoter drives transcription of Nat4 at similar levels to the native NAT4 promoter and, importantly, its activity is not affected by CR (see revised Fig 1A). Using this newly constructed yeast strain (Pste5-NAT4), we show in new figures 1B and 6E that constitutive expression of Nat4

prevents the reduction of N-acH4 levels under CR compared to a WT strain (compare green and blue bars). Furthermore, these figures show that CR does not further impact on the N-acH4 levels in *nat4Δ* mutant compared to WT cells (Figs 1B and 6E, compare red and blue bars in CR) suggesting that CR impacts on this histone modification mainly through Nat4.

The authors have addressed the reviewers' concerns regarding the role of *nat4* in longevity by observing the effect of constitutive expression of Nat4 on N-acH4 under calorie restriction conditions. To ensure that the effects are clear to the reader, they should include statistical significance tests that are represented in the new figures (such as the asterisks representing p-values as they have done for other figures). Currently, it is unclear in Fig 1B whether the effect of constitutive expression of Nat4 is specific to CR conditions, as it also seems that the *Pste5-NAT4* has a similar effect in NCR compared to WT (see blue and green bars in NCR compared to blue and green bars in CR). The effects do appear to be more clear in Fig 6B, however statistics would also be appreciated here. In case the prevention of the reduction of N-acH4 in the constitutively expressed Nat4 strain is not specific to CR, the authors should address this in the body of the paper, as it does not appear to be due to a difference in expression level of Nat4 between the two strains (see blue bars in Fig 1A).

The authors have also analyzed the effect of Nat4 deletion on N-acH4 under CR conditions. The results here are clearer, that CR does not further affect N-acH4 in the Nat4 deletion, which is in contrast to NCR conditions. Again, including statistical significance in the figure would be helpful to the reader.

In figure 1F of the revised manuscript we show that constitutive expression of Nat4 in the *Pste5-NAT4* strain limits the extension of lifespan mediated by CR compared to the longevity observed in a WT strain (from 32% down to 19%). This result supports the conclusion of this study presented in figure 7, which demonstrates that Nat4 functions within one of the pathways induced by CR to regulate cellular lifespan.

This is a nice addition to the manuscript, and the results are clear. The authors have sufficiently addressed whether constitutive Nat4 expression prevents lifespan extension by CR.

While we agree with the referees' point that the connection between *Pnc1* and *nat4Δ*-induced longevity needs further exploration, we believe that overexpressing *Pnc1* in *nat4Δ* cells would not provide further information on this issue because *Pnc1* is already overexpressed in *nat4Δ* mutants (Figs 3C, 4C, 4D, 6A and Table EV1). Alternatively, we examined lifespan in *nat4Δ* cells lacking the transcription factors *Msn2* and *Msn4* which directly activate the expression of *Pnc1* (Medvedik et al, PLoS Biology 2007) and we show that the triple mutant *msn2Δ msn4Δ nat4Δ* is no longer able to extend lifespan (Fig 6C). This is another evidence reinforcing the idea that *Pnc1* induction is required for the extension of lifespan in *nat4Δ* cells.

The fact that the triple mutant *msn2Δ msn4Δ nat4Δ* is no longer able to extend lifespan presumably because *Pnc1* is no longer induced is a nice addition to the manuscript. However, this does not directly address the request for the experiment with *Pnc1* overexpression in the Nat4 mutant. The authors rebuttal appears to be satisfactory, that the effect of *Pnc1* overexpression was not tested in *nat4Δ* because it is already overexpressed in these cells and also because *Pnc1* overexpression alone results in increased lifespan. The *Pnc1* connection could still be further strengthened by including some evidence that *Pnc1* is not induced in the *msn2Δ msn4Δ nat4Δ* in comparison to *nat4Δ* cells by RT-qPCR for example.

Finally, we examined lifespan in *sir2Δ fob1Δ* cells overexpressing *Pnc1* and found that lifespan is not extended indicating that *Pnc1*-mediated longevity is dependent on Sir2 as previously described (Anderson et al., Nature 2003). We also examined lifespan in *sir2Δ fob1Δ* cells lacking Nat4 and found that *nat4Δ* cannot extend lifespan in this mutant background, resembling the effect of *Pnc1* above. Therefore, this further verifies that Nat4 and *Pnc1* function within the same pathway. These additional data are described in the results section and shown in figures 6D and EV6.

The experiments using *sir2Δ fob1Δ* cells are a good addition to the manuscript, particularly that neither *nat4* deletion nor overexpression of *Pnc1* does not increase lifespan in these cells.

Referee 2 also indicates in her/his cross-comments that point 2 of referee 1 would not need to be addressed experimentally. The lifespan curves should be repeated and remaining variations explained. Point 5 of referee 1 does also not need to be addressed. Referee 2 further feels that addressing points 3 and 4 of referee 3 would not be strictly required for publication of the paper here.

Lifespan curves have been obtained after analyzing a minimum of 60 independent cells for each mutant, and examining in parallel WT control cells during each RLS experiment. The mean lifespan numbers obtained for each of the various yeast genetic background strains used have been consistently reproducible during the implementation of this study. We performed additional lifespan experiments for *pnc1Δ* strains as recommended and we now show in revised figure 6B that all strains tested have similarly shaped survival curves. As in the previous version of the manuscript, these results show that loss of Pnc1 prevents *nat4Δ*-mediated longevity.

This point has been sufficiently addressed.

Referee #2:

Reviewer comments:

Overall this is a very nice and clear manuscript and a further demonstration of the links between environmental signals that are incorporated as histone modifications to affect cellular aging. The finding that *nat4Δ* is epistatic to calorie restriction-induced longevity is exciting. Given the quality and scope of the manuscript, if the detailed comments below can be addressed, the paper is suitable for publication in EMBO Reports.

1. The authors claim in Figure 1A that NAT4 expression decreases with calorie restriction. While the NAT4 RNA levels have indeed been shown to decrease, it would be more convincing if the authors could show that the level of NAT4 protein is also decreasing with CR, e.g. by western blot.

We have attempted the experiment suggested by the referee but the available antibodies, which have been raised against the human ortholog (Naa40), cannot detect yeast Nat4 (Fig EV5). Therefore, we have also HA-tagged the endogenous Nat4 but again we could not detect the protein with the HA antibody (data not shown) because Nat4 is expressed at very low levels. Despite the inability to detect Nat4 protein, we believe the finding showing that Nat4 RNA levels decrease under CR (Fig 1A and Table EV3) is supplemented by the reduction of the Nat4-catalysed modification N-acH4 in CR (Fig 1B and 6E).

This point has been sufficiently addressed.

2. Figure 1E is missing *nat4Δ* NCR.

We have now performed additional RLS experiments for all strains in this figure and have included the data on *nat4Δ* NCR in revised figure 1E.

This point has been sufficiently addressed.

3. In Figure 2A please explain abbreviations (NTS, ETS, ITS etc) either directly in the figure or in the figure legend.

We thank the referee for spotting this. We have now explained these abbreviations in the figure legend.

This point has been sufficiently addressed.

4. On page 6 of the manuscript and in Figure 2D, the authors claim that ribosome biogenesis is not affected in *nat4Δ*. However, in the RNA-seq data in the subsequent results section on page 7 and Figure 3A ribonucleoprotein biogenesis is shown to be downregulated. Though this contradiction is addressed in the discussion on page 12, it is confusing when reading the results section. Some extra clarifications in the results section would be helpful.

Following the referee's suggestion we clarified this contradiction in the results section. Additionally, in the title of the section referring to figure 2D we replaced the words "ribosome biogenesis" with "polysome profiling" for further clarification.

This point has been sufficiently addressed.

5. While the experiments in the manuscript have shown the effect of deleting Nat4 and loss of N-acH4 on lifespan, it would be interesting to know the effects of overexpressing Nat4. Specific questions that could be addressed are:

What is the effect of overexpressing Nat4 on N-acH4 level in the rDNA region?

How does overexpression of Nat4 impact lifespan?

We have generated a new strain (Pste5-NAT4) in which Nat4 is constitutively expressed at physiological levels even under CR conditions (revised Fig 1A). Using this strain we now show that constitutive Nat4 expression maintains high N-acH4 levels under CR (Figs 1B and 6E), reduces the induction of stress response genes in CR (Fig 3D), and limits lifespan extension (from 32% down to 19%) mediated by CR (Fig 1F).

The effects of constitutive expression of Nat4 during CR and the maintenance of high NacH4 levels are interesting and add significantly to the manuscript. The use of a promoter that expresses similar to the native level of Nat4 under NCR conditions was a good choice and the results are satisfactory.

Cross-comments by referee 2 on referee 1's report:

The authors have added some important experimental evidence in response to the major referee comments. I am still concerned, however, that the manuscript is structured awkwardly and fails to make a convincing and clear case for the model that the authors propose. In particular, the sir2 fob1 data while an important addition is also difficult to interpret, since the authors argue that nat4 deletion mimics CR, yet several groups have published that CR increases lifespan more robustly in the sir2 fob1 double mutant compared to WT, as does deletion of tor1. Does CR still extend lifespan of the sir2 fob1 nat4 triple mutant? Either Nat4 is in the same pathway with CR and TOR or it isn't. The msn2 msn4 data are also not consistent with prior results where this double deletion mutant is long-lived. The PSY316 strain should be avoided for these studies, as activation of Sir2 fails to extend lifespan in that strain, a major flaw with some of the prior literature on Pnc1.

The suggestion to change the structure of the paper is not necessary, it should be up to the authors/editor.

After examining literature on the discrepancies between the PSY316 and BY4741 strains, it appears that the Reviewer #1 has some valid points. The authors have used the PSY316 background strain to show that overexpression of Pnc1 does not result in an increase in sir2delta/fob1delta (Fig EV6). This shows as they suggest in the manuscript, that Pnc1-mediated longevity is dependent on Sir2. Unfortunately the authors compare 2 background strains that appear to have different results when Sir2 is overexpressed. This is not necessarily a problem as far as in the results they state that both Pnc1 and Nat4 appear to depend on Sir2. However, in the discussion and their model in Fig 7, they suggest a possible mechanism where increased Pnc1 expression leads to activation of Sir2 and this in turn leads to lifespan extension. Unfortunately in the PSY316 strain, overexpression of Sir2 does not result in lifespan extension (Kaeberlein et al 2004). If this is robust, it could mean that lifespan extension depends on Sir2 but is not due to increased activity of Sir2, or it could mean that somehow overexpression of Sir2 does not resemble the effect of Sir2 stimulation. The only experiment that they used this PSY316 background is for the overexpression experiment shown in Fig EV6, thus for the majority of the experiments in the manuscript, there are no strain discrepancies. Here it is up to the reviewer/editor to decide whether explicitly stating this in the manuscript is sufficient, or whether there needs to be further experiments e.g. repeating the Pnc1 overexpression in the BY4741 background.

I think that deletion of Nat4 in the sir2Δ fob1Δ was a nice experiment to show that Nat4 is dependent on Sir2 for lifespan extension in Fig 6D. Whether or not lifespan is further extended by

CR is an independent question here. As is well known and also explicitly shown in the model in Fig 7, CR can extend lifespan via multiple pathways. I think the point the authors were making has been made satisfactorily.

Regarding the reviewer's comment that *msn2Δ msn4Δ* data are not consistent with prior results where this double deletion mutant is long-lived, I have seen conflicting evidence in the literature where sometimes this double mutant already shows lifespan extension while sometimes it does not. However, I think that the point that *nat4Δ* can no longer extend lifespan in this background is valid based on the authors' data in Fig 6C.

I also do not accept the authors' explanation in the rebuttal to my question about the glucose concentration in the culture medium. The fact that the cells were harvested at an OD of 0.8, which the authors claim is still log-phase for cells initiated in 0.1% glucose, suggests a possible lack of experimental rigor. I find it hard to believe that cells started in a 0.1% glucose culture will not have exhausted nearly all of the glucose by the time they reach OD of 0.8, while cells started in 2% glucose will certainly have exhausted a significant portion of the glucose. This can and should be quantified and experimentally corrected if necessary.

The authors have stated the following in their rebuttal: "We note in Materials & Methods that yeast cultures for every experiment were initiated at dilution O.D. 0.1 with media containing the indicated glucose concentration (2% or 0.1%) and then cells were harvested during the exponential growth phase at O.D. 0.8. At this O.D. value cells are still in growth phase and therefore have not exhausted glucose from the media."

This suggests that the authors believe that the cells grown in 0.1% glucose to OD 0.8 are still in log/growth phase and therefore glucose exhaustion is not an important consideration. However Reviewer #1 is not satisfied with this rebuttal because the authors have not shown it. Though it is well accepted that yeast are still in growth phase at OD 0.8 when grown in 2% glucose, it is unclear from the authors' rebuttal whether they have tested that these cells are still in log phase at OD 0.8 when grown in 0.1% glucose. It is a simple experiment to do. If it happens that these cells are shown to be in log phase, then the point will have been addressed as satisfactorily as possible. If these cells are not in log phase, however, it is up to the reviewer/editor to decide whether it is enough to explicitly state in the manuscript that glucose exhaustion may have an effect or whether it is necessary to repeat the expression experiments with cells in log phase.

However, note that in the original review, Reviewer #1 had said that this was a minor point and written as follows: "It's not clear from the methods how much glucose is actually present in the cultures at the times that the assays for gene expression are performed. How much glucose is used up in outgrowth to "mid-log phase" under these conditions? This is important both for interpretation of the meaning of the data, but also so that others will be able to replicate these findings."

Based on this comment, it was not clear that the authors should experimentally show how much glucose is consumed in their experiments, so it is up to the reviewer/editor to decide whether this is necessary for publication of the manuscript.

2nd Revision - authors' response

21 September 2016

Response to comments

Thank you for the submission of your revised manuscript to our journal. We have now received the comments from referees 1 and 2; referee 3 was unfortunately not available to assess the revised manuscript for us. I have asked referee 2 to please also examine how well referee 3's concerns were addressed.

As you will see, while the referees acknowledge that the study has been strengthened and improved, both also still have remaining concerns that need to be taken into account. These are the major 4 points/suggestions:

- Pnc1 overexpression in the BY4741 background

We appreciate the concern raised regarding the use of the PSY316 strain. However, we believe that the use of this strain in one of the lifespan experiments presented (Fig EV4B) does not diminish the overall conclusion of this paper that depletion of Nat4 partially mimics calorie-restriction to induce the expression of Pnc1 and mediate longevity in a Sir2-dependent manner. Nevertheless, we discuss the discrepancy of using the PSY316 strain for a lifespan experiment within the text and provide potential explanations for the obtained result. Specifically, we argue that overexpression of Sir2 in the PSY316 background strain may not recapitulate the longevity effects mediated by increased Sir2 activity. This is also supported by our RNA-seq data which show that Sir2 is not overexpressed in nat4-delta cells, but lifespan is still extended in these cells through induction of Pnc1 and is dependent on Sir2 (Fig 6D).

- show that the cells grown in 0.1% glucose to OD 0.8 are still in log/growth phase

We performed growth curves for BY4741 cells cultured in 2% and 0.1% glucose and show that in both conditions the cells at O.D. 0.8 have not exhausted glucose (as they have not passed through the diauxic shift) and thus are still in log phase.

- demonstrate statistical significance in the new figures (this needs to be done for all figures!)

We performed statistical analysis for all figures in this paper and demonstrate the required statistical significance.

- show that Pnc1 is not induced in the msn2Δ msn4Δ nat4Δ strain in comparison to nat4Δ cells

We performed RT-qPCR to show that PNC1 is not induced in the triple msn2Δ msn4Δ nat4Δ mutant strain in comparison to nat4Δ cells (Fig EV4A). In fact, Pnc1 is downregulated in the triple mutant compared to WT cells as previously reported (Medvedik et al, PLoS Biol 2007).

I think that all 4 points should be addressed experimentally. If point 1 cannot be done, please discuss the discrepancy with earlier studies and provide potential explanations. I would therefore like to give you the opportunity to address the remaining concerns in a second revision that should be performed and the final manuscript submitted as soon as possible.

The manuscript currently has 7 main plus 7 EV figures. Unfortunately, at the moment, we can only offer 5 EV figures per manuscript. However, your EV figures are predominantly single panels only, and could potentially be combined into a total of 5 EV figures.

We have consolidated our supplementary data into 5 EV figures. We have also deposited our transcriptome data in GEO and provide the accession number in the Materials and Methods section.

EMBO press papers are accompanied online by A) a short (1-2 sentences) summary of the findings and their significance, B) 2-3 bullet points highlighting key results and C) a synopsis image that is 550x200-400 pixels large (the height is variable). You can either show a model or key data in the synopsis image. Please note that text needs to be readable at the final size. Please send us this information along with the revised manuscript.

We have sent this information along with the revised manuscript.

Referee #1:

The authors have added some important experimental evidence in response to the major referee comments. I am still concerned, however, that the manuscript is structured awkwardly and fails to

make a convincing and clear case for the model that the authors propose. In particular, the sir2 fob1 data while an important addition is also difficult to interpret, since the authors argue that nat4 deletion mimics CR, yet several groups have published that CR increases lifespan more robustly in the sir2 fob1 double mutant compared to WT, as does deletion of tor1. Does CR still extend lifespan of the sir2 fob1 nat4 triple mutant? Either Nat4 is in the same pathway with CR and TOR or it isn't. The msn2 msn4 data are also not consistent with prior results where this double deletion mutant is long-lived. The PSY316 strain should be avoided for these studies, as activation of Sir2 fails to extend lifespan in that strain, a major flaw with some of the prior literature on Pnc1.

We thank the referee for raising these concerns and the insightful comments. We beg to differ regarding the structure of the paper because we believe that it is important to emphasize the rDNA-related results.

According to our results, we claim in the manuscript that Nat4 functions downstream of one of the many CR-mediated pathways (Fig 7). Therefore, it remains possible that CR increases lifespan in the sir2 Δ fob1 Δ double mutant through a Nat4-independent manner. The referee's question of whether CR affects lifespan in the sir2 Δ fob1 Δ nat4 Δ triple mutant is very interesting but independent to the work presented here and therefore, beyond the scope of this manuscript.

Our data on msn2 Δ msn4 Δ lifespan are consistent with previous reports including Medvedik et al. PLoS Biology 2007 and Wei et al., PLoS Genetics 2008. Perhaps conflicting results on the lifespan of this double mutant appear within the literature due to the use of different genetic backgrounds, culture conditions etc. Despite this, our main conclusion derived from this experiment that nat4-delta can no longer extend lifespan in the msn2 Δ msn4 Δ double mutant (Fig 6C) still stands regardless of the msn2 Δ msn4 Δ longevity effect.

We appreciate the concern regarding the use of the PSY316 strain in the lifespan assay shown in figure EV4B. We first alert the readers about a possible discrepancy with previous studies and then we provide a plausible scenario that Sir2 overexpression might not reflect the longevity effect mediated by Sir2 activation. This argument is supported by the RNAseq data (Fig 3A and Supplementary tables) from this work showing that in nat4-delta cells Sir2 is not overexpressed even though in these cells Pnc1 is induced and lifespan is extended in a sir2-dependent manner. Hence, we suggest through our model that nat4-mediated longevity requires Sir2 but it is not dependent on its overexpression.

I also do not accept the authors' explanation in the rebuttal to my question about the glucose concentration in the culture medium. The fact that the cells were harvested at an OD of 0.8, which the authors claim is still log-phase for cells initiated in 0.1% glucose, suggests a possible lack of experimental rigor. I find it hard to believe that cells started in a 0.1% glucose culture will not have exhausted nearly all of the glucose by the time they reach OD of 0.8, while cells started in 2% glucose will certainly have exhausted a significant portion of the glucose. This can and should be quantified and experimentally corrected if necessary.

Following the referee's suggestion we addressed this concern by generating growth curves for BY4741 cells cultured in 2% and 0.1% glucose. We found that cells in 0.1% glucose are indeed growing less overall than cells in 2% glucose but, at O.D. 0.8 cells in both glucose concentrations grow at a similar rate, are still in growth phase and do not enter stationary phase (Fig EV1A).

Referee #2:

Response to Author rebuttal

Briefly, the authors have nicely addressed the concerns and requests of the reviewers. For publication in EMBO reports, I would suggest that the authors include statistical significance in their new figures as well as showing that Pnc1 is not induced in the msn2 Δ msn4 Δ nat4 Δ strain in comparison to nat4 Δ cells by RT-qPCR for example. Please see the point-by-point responses below.

Following the referees suggestions we have included statistical significance to all the figures within the manuscript and have compared the expression of *PNC1* in the triple mutant *msn2Δ msn4Δ nat4Δ* and single mutant *nat4Δ* strains by RT-qPCR (Fig EV4A).

Dear Antonis,

Thank you for the submission of your research manuscript to EMBO reports. We have now received the enclosed referee reports on your study that are copied below, as well as referee cross-comments. As you will see, the referees acknowledge that the findings are potentially interesting. However, they also suggest some more experiments that would be required to strengthen the study. On one hand, more data are needed to confirm that Nat4 regulates lifespan, and on the other hand, it remains somewhat unclear how Pnc1 affects longevity. While strengthening both aspects would be most welcome, establishing a clear role for Nat4 as lifespan regulator downstream of CR is more important for publication of the paper here. Referee 2 agrees with referee 3 (in the cross-comments) that it should be analyzed whether constitutive Nat4 expression maintains high N-acH4 levels under CR, and whether CR impacts on N-acH4 levels in a Nat4 deletion mutant. Whether constitutive Nat4 expression prevents lifespan extension by CR should also be examined. Both referees further agree that it should be tested how Pnc1 overexpression affects lifespan in *nat4Δ* and *sir2/fob1Δ* cells. These experiments should therefore be performed during revision of the manuscript for EMBO reports.

To address the points that have been raised by referees we have constructed a new yeast strain in which the expression of Nat4 is driven by the constitutive STE5 promoter (Pste5-NAT4). This promoter drives transcription of Nat4 at similar levels to the native NAT4 promoter and, importantly, its activity is not affected by CR (see revised Fig 1A). Using this newly constructed yeast strain (Pste5-NAT4), we show in new figures 1B and 6E that constitutive expression of Nat4 prevents the reduction of N-acH4 levels under CR compared to a WT strain (compare green and blue bars). Furthermore, these figures show that CR does not further impact on the N-acH4 levels in *nat4Δ* mutant compared to WT cells (Figs 1B and 6E, compare red and blue bars in CR) suggesting that CR impacts on this histone modification mainly through Nat4.

The authors have addressed the reviewers' concerns regarding the role of *nat4* in longevity by observing the effect of constitutive expression of Nat4 on N-acH4 under calorie restriction conditions. To ensure that the effects are clear to the reader, they should include statistical significance tests that are represented in the new figures (such as the asterisks representing p-values as they have done for other figures). Currently, it is unclear in Fig 1B whether the effect of constitutive expression of Nat4 is specific to CR conditions, as it also seems that the Pste5-NAT4 has a similar effect in NCR compared to WT (see blue and green bars in NCR compared to blue and green bars in CR). The effects do appear to be more clear in Fig 6B, however statistics would also be appreciated here. In case the prevention of the reduction of N-acH4 in the constitutively expressed Nat4 strain is not specific to CR, the authors should address this in the body of the paper, as it does not appear to be due to a difference in expression level of Nat4 between the two strains (see blue bars in Fig 1A).

We have now performed statistical analysis on all new figures including the ones mentioned by the referee above. We show by statistics that the effect of Nat4 constitutive expression (Pste5-Nat4) on the levels of N-acH4 is specific to CR conditions at most rDNA loci (Fig 1B) and this observation is even more robust along the *PNC1* gene in fig 6E.

The authors have also analyzed the effect of Nat4 deletion on N-acH4 under CR conditions. The results here are clearer, that CR does not further affect N-acH4 in the Nat4 deletion, which is in contrast to NCR conditions. Again, including statistical significance in the figure would be helpful to the reader.

We performed the required statistical analysis and found no significant difference on the effect of *nat4* deletion on N-acH4 levels between CR and NCR conditions. We describe this finding within the results section.

In figure 1F of the revised manuscript we show that constitutive expression of Nat4 in the Pste5-NAT4 strain limits the extension of lifespan mediated by CR compared to the longevity observed in a WT strain (from 32% down to 19%). This result supports the conclusion of this study presented in figure 7, which demonstrates that Nat4 functions within one of the pathways induced by CR to regulate cellular lifespan.

This is a nice addition to the manuscript, and the results are clear. The authors have sufficiently addressed whether constitutive Nat4 expression prevents lifespan extension by CR.

While we agree with the referees' point that the connection between Pnc1 and nat4 Δ -induced longevity needs further exploration, we believe that overexpressing Pnc1 in nat4 Δ cells would not provide further information on this issue because Pnc1 is already overexpressed in nat4 Δ mutants (Figs 3C, 4C, 4D, 6A and Table EV1). Alternatively, we examined lifespan in nat4 Δ cells lacking the transcription factors Msn2 and Msn4 which directly activate the expression of Pnc1 (Medvedik et al, PLoS Biology 2007) and we show that the triple mutant msn2 Δ msn4 Δ nat4 Δ is no longer able to extend lifespan (Fig 6C). This is another evidence reinforcing the idea that Pnc1 induction is required for the extension of lifespan in nat4 Δ cells.

The fact that the triple mutant msn2 Δ msn4 Δ nat4 Δ is no longer able to extend lifespan presumably because Pnc1 is no longer induced is a nice addition to the manuscript. However, this does not directly address the request for the experiment with Pnc1 overexpression in the Nat4 mutant. The authors rebuttal appears to be satisfactory, that the effect of Pnc1 overexpression was not tested in nat4 delta because it is already overexpressed in these cells and also because Pnc1 overexpression alone results in increased lifespan. The Pnc1 connection could still be further strengthened by including some evidence that Pnc1 is not induced in the msn2 Δ msn4 Δ nat4 Δ in comparison to nat4 Δ cells by RT-qPCR for example.

We performed RT-qPCR analysis in msn2 Δ msn4 Δ nat4 Δ strain, as well as in WT, nat4 Δ and the msn2 Δ msn4 Δ double mutant strain (Fig EV4A). In agreement to our previous findings, Pnc1 is upregulated in the nat4-delta strain in comparison to WT cells. In contrast, *PNC1* is downregulated in the double mutant msn2 Δ msn4 Δ in comparison to the WT strain which is consistent with the decrease in Pnc1 protein levels reported previously (Medvedik et al., PLoS Biology, 2007). Importantly, the mRNA levels of *PNC1* remain reduced in the msn2 Δ msn4 Δ nat4 Δ triple mutant strain in comparison to WT cells.

Finally, we examined lifespan in sir2 Δ fob1 Δ cells overexpressing Pnc1 and found that lifespan is not extended indicating that Pnc1-mediated longevity is dependent on Sir2 as previously described (Anderson et al., Nature 2003). We also examined lifespan in sir2 Δ fob1 Δ cells lacking Nat4 and found that nat4 Δ cannot extend lifespan in this mutant background, resembling the effect of Pnc1 above. Therefore, this further verifies that Nat4 and Pnc1 function within the same pathway. These additional data are described in the results section and shown in figures 6D and EV6.

The experiments using sir2 Δ fob1 Δ cells are a good addition to the manuscript, particularly that neither nat4 deletion nor overexpression of Pnc1 does not increase lifespan in these cells.

Referee 2 also indicates in her/his cross-comments that point 2 of referee 1 would not need to be addressed experimentally. The lifespan curves should be repeated and remaining variations explained. Point 5 of referee 1 does also not need to be addressed. Referee 2 further feels that addressing points 3 and 4 of referee 3 would not be strictly required for publication of the paper here.

Lifespan curves have been obtained after analyzing a minimum of 60 independent cells for each mutant, and examining in parallel WT control cells during each RLS experiment. The mean lifespan numbers obtained for each of the various yeast genetic background strains used have been consistently reproducible during the implementation of this study. We performed additional lifespan experiments for pnc1 Δ strains as recommended and we now show in revised figure 6B that all strains tested have similarly shaped survival curves. As in the previous version of the manuscript, these results show that loss of Pnc1 prevents nat4 Δ -mediated longevity.

This point has been sufficiently addressed.

Referee #2:

Reviewer comments:

Overall this is a very nice and clear manuscript and a further demonstration of the links between environmental signals that are incorporated as histone modifications to affect cellular aging. The finding that *nat4Δ* is epistatic to calorie restriction-induced longevity is exciting. Given the quality and scope of the manuscript, if the detailed comments below can be addressed, the paper is suitable for publication in EMBO Reports.

1. The authors claim in Figure 1A that NAT4 expression decreases with calorie restriction. While the NAT4 RNA levels have indeed been shown to decrease, it would be more convincing if the authors could show that the level of NAT4 protein is also decreasing with CR, e.g. by western blot.

We have attempted the experiment suggested by the referee but the available antibodies, which have been raised against the human ortholog (Naa40), cannot detect yeast Nat4 (Fig EV5). Therefore, we have also HA-tagged the endogenous Nat4 but again we could not detect the protein with the HA antibody (data not shown) because Nat4 is expressed at very low levels. Despite the inability to detect Nat4 protein, we believe the finding showing that Nat4 RNA levels decrease under CR (Fig 1A and Table EV3) is supplemented by the reduction of the Nat4-catalysed modification N-acH4 in CR (Fig 1B and 6E).

This point has been sufficiently addressed.

2. Figure 1E is missing *nat4Δ* NCR.

We have now performed additional RLS experiments for all strains in this figure and have included the data on *nat4Δ* NCR in revised figure 1E.

This point has been sufficiently addressed.

3. In Figure 2A please explain abbreviations (NTS, ETS, ITS etc) either directly in the figure or in the figure legend.

We thank the referee for spotting this. We have now explained these abbreviations in the figure legend.

This point has been sufficiently addressed.

4. On page 6 of the manuscript and in Figure 2D, the authors claim that ribosome biogenesis is not affected in *nat4Δ*. However, in the RNA-seq data in the subsequent results section on page 7 and Figure 3A ribonucleoprotein biogenesis is shown to be downregulated. Though this contradiction is addressed in the discussion on page 12, it is confusing when reading the results section. Some extra clarifications in the results section would be helpful.

Following the referee's suggestion we clarified this contradiction in the results section. Additionally, in the title of the section referring to figure 2D we replaced the words "ribosome biogenesis" with "polysome profiling" for further clarification.

This point has been sufficiently addressed.

5. While the experiments in the manuscript have shown the effect of deleting Nat4 and loss of N-acH4 on lifespan, it would be interesting to know the effects of overexpressing Nat4. Specific questions that could be addressed are:

What is the effect of overexpressing Nat4 on N-acH4 level in the rDNA region?
How does overexpression of Nat4 impact lifespan?

We have generated a new strain (Pste5-NAT4) in which Nat4 is constitutively expressed at physiological levels even under CR conditions (revised Fig 1A). Using this strain we now show that

constitutive Nat4 expression maintains high N-acH4 levels under CR (Figs 1B and 6E), reduces the induction of stress response genes in CR (Fig 3D), and limits lifespan extension (from 32% down to 19%) mediated by CR (Fig 1F).

The effects of constitutive expression of Nat4 during CR and the maintenance of high NacH4 levels are interesting and add significantly to the manuscript. The use of a promoter that expresses similar to the native level of Nat4 under NCR conditions was a good choice and the results are satisfactory.

Cross-comments by referee 2 on referee 1's report:

The authors have added some important experimental evidence in response to the major referee comments. I am still concerned, however, that the manuscript is structured awkwardly and fails to make a convincing and clear case for the model that the authors propose. In particular, the sir2 fob1 deletion mimics CR, yet several groups have published that CR increases lifespan more robustly in the sir2 fob1 double mutant compared to WT, as does deletion of tor1. Does CR still extend lifespan of the sir2 fob1 nat4 triple mutant? Either Nat4 is in the same pathway with CR and TOR or it isn't. The msn2 msn4 data are also not consistent with prior results where this double deletion mutant is long-lived. The PSY316 strain should be avoided for these studies, as activation of Sir2 fails to extend lifespan in that strain, a major flaw with some of the prior literature on Pnc1.

The suggestion to change the structure of the paper is not necessary, it should be up to the authors/editor.

After examining literature on the discrepancies between the PSY316 and BY4741 strains, it appears that the Reviewer #1 has some valid points. The authors have used the PSY316 background strain to show that overexpression of Pnc1 does not result in an increase in sir2delta/fob1delta (Fig EV6). This shows as they suggest in the manuscript, that Pnc1-mediated longevity is dependent on Sir2. Unfortunately the authors compare 2 background strains that appear to have different results when Sir2 is overexpressed. This is not necessarily a problem as far as in the results they state that both Pnc1 and Nat4 appear to depend on Sir2. However, in the discussion and their model in Fig 7, they suggest a possible mechanism where increased Pnc1 expression leads to activation of Sir2 and this in turn leads to lifespan extension. Unfortunately in the PSY316 strain, overexpression of Sir2 does not result in lifespan extension (Kaeberlein et al 2004). If this is robust, it could mean that lifespan extension depends on Sir2 but is not due to increased activity of Sir2, or it could mean that somehow overexpression of Sir2 does not resemble the effect of Sir2 stimulation. The only experiment that they used this PSY316 background is for the overexpression experiment shown in Fig EV6, thus for the majority of the experiments in the manuscript, there are no strain discrepancies. Here it is up to the reviewer/editor to decide whether explicitly stating this in the manuscript is sufficient, or whether there needs to be further experiments e.g. repeating the Pnc1 overexpression in the BY4741 background.

Our response to this concern is provided above.

I think that deletion of Nat4 in the sir2 Δ fob1 Δ was a nice experiment to show that Nat4 is dependent on Sir2 for lifespan extension in Fig 6D. Whether or not lifespan is further extended by CR is an independent question here. As is well known and also explicitly shown in the model in Fig 7, CR can extend lifespan via multiple pathways. I think the point the authors were making has been made satisfactorily.

Regarding the reviewer's comment that msn2 Δ msn4 Δ data are not consistent with prior results where this double deletion mutant is long-lived, I have seen conflicting evidence in the literature where sometimes this double mutant already shows lifespan extension while sometimes it does not. However, I think that the point that nat4 Δ can no longer extend lifespan in this background is valid based on the authors' data in Fig 6C.

I also do not accept the authors' explanation in the rebuttal to my question about the glucose concentration in the culture medium. The fact that the cells were harvested at an OD of 0.8, which the authors claim is still log-phase for cells initiated in 0.1% glucose, suggests a possible lack of

experimental rigor. I find it hard to believe that cells started in a 0.1% glucose culture will not have exhausted nearly all of the glucose by the time they reach OD of 0.8, while cells started in 2% glucose will certainly have exhausted a significant portion of the glucose. This can and should be quantified and experimentally corrected if necessary.

The authors have stated the following in their rebuttal: "We note in Materials & Methods that yeast cultures for every experiment were initiated at dilution O.D. 0.1 with media containing the indicated glucose concentration (2% or 0.1%) and then cells were harvested during the exponential growth phase at O.D. 0.8. At this O.D. value cells are still in growth phase and therefore have not exhausted glucose from the media."

This suggests that the authors believe that the cells grown in 0.1% glucose to OD 0.8 are still in log/growth phase and therefore glucose exhaustion is not an important consideration. However Reviewer #1 is not satisfied with this rebuttal because the authors have not shown it. Though it is well accepted that yeast are still in growth phase at OD 0.8 when grown in 2% glucose, it is unclear from the authors' rebuttal whether they have tested that these cells are still in log phase at OD 0.8 when grown in 0.1% glucose. It is a simple experiment to do. If it happens that these cells are shown to be in log phase, then the point will have been addressed as satisfactorily as possible. If these cells are not in log phase, however, it is up to the reviewer/editor to decide whether it is enough to explicitly state in the manuscript that glucose exhaustion may have an effect or whether it is necessary to repeat the expression experiments with cells in log phase.

However, note that in the original review, Reviewer #1 had said that this was a minor point and written as follows: "It's not clear from the methods how much glucose is actually present in the cultures at the times that the assays for gene expression are performed. How much glucose is used up in outgrowth to "mid-log phase" under these conditions? This is important both for interpretation of the meaning of the data, but also so that others will be able to replicate these findings."

Based on this comment, it was not clear that the authors should experimentally show how much glucose is consumed in their experiments, so it is up to the reviewer/editor to decide whether this is necessary for publication of the manuscript.

The response to this concern is provided above.

Accepted

30 September 2016

I am very pleased to accept your manuscript for publication in the next available issue of EMBO reports. Thank you for your contribution to our journal.

Corresponding Author Name: Antonis Kirmizis

Manuscript Number: EMBOR-2016-42540